# Independently engaging protein tethers of different length enhance synaptic vesicle trafficking to the plasma membrane

Manon M. M. Berns[1] 🔵 , Mirza Yildiz[2], Stefanie Winkelmann[2] 🔵 and Alexander M. Walter[1] 🔵

[1]*Department of Neuroscience, University of Copenhagen, Copenhagen, Denmark*
[2]*Zuse Institute Berlin, Berlin, Germany*

Handling Editors: Katalin Toth & Samuel Young

The peer review history is available in the Supporting Information section of this article (https://doi.org/10.1113/JP286651#support-information-section).

**Abstract figure legend** The independent assembly of the protein tethers synaptotagmin (Syt1/2), Munc13 and the SNAREs (upper) affects the stochastic movement of a synaptic vesicle (SV) with respect to the plasma membrane (lower). SV and proteins are not drawn to scale. Proteins are only depicted in the illustration once they are tethering the SV to the plasma membrane (PM).

**Manon Berns** is a neuroscientist with a profound interest in mathematical modelling. She studied both Neurosciences and Bioinformatics & Systems Biology at the Vrije Universiteit Amsterdam. She joined Alexander Walter's lab at the University of Copenhagen and obtained her PhD in June 2024. Her research focuses on understanding the molecular mechanisms of neurotransmitter release. She combines available knowledge and information into mathematical frameworks, thereby aiming to untangle the complex dynamics of synaptic vesicle docking, priming and fusion.

The Journal of Physiology

**Abstract** Synaptic vesicle (SV) trafficking toward the plasma membrane (PM) and subsequent SV maturation are essential for neurotransmitter release. These processes, including SV docking and priming, are co-ordinated by various proteins, such as SNAREs, Munc13 and synaptotagmin (Syt), which connect (tether) the SV to the PM. Here, we investigated how tethers of varying lengths mediate SV docking using a simplified mathematical model. The heights of the three tether types, as estimated from the structures of the SNARE complex, Munc13 and Syt, defined the SV–PM distance ranges for tether formation. Geometric considerations linked SV–PM distances to the probability and rate of tether formation. We assumed that SV tethering constrains SV motility and that multiple tethers are associated by independent interactions. The model predicted that forming multiple tethers favours shorter SV–PM distances. Although tethers acted independently in the model, their geometrical properties often caused their sequential assembly, from longer ones (Munc13/Syt), which accelerated SV movement towards the PM, to shorter ones (SNAREs), which stabilized PM-proximal SVs. Modifying tether lengths or numbers affected SV trafficking. The independent implementation of tethering proteins enabled their selective removal to mimic gene knockout (KO) situations. This showed that simulated SV–PM distance distributions qualitatively aligned with published electron microscopy studies upon removal of SNARE and Syt tethers, whereas Munc13 KO data were best approximated when assuming additional disruption of SNARE tethers. Thus, although salient features of SV docking can be accounted for by independent tethering alone, our results suggest that functional tether interactions not yet featured in our model are crucial for biological function.

(Received 3 April 2024; accepted after revision 18 December 2024; first published online 14 January 2025)

**Corresponding authors** M. M. M. Berns and A. M. Walter: Department of Neuroscience, University of Copenhagen, Blegdamsvej 3B, 2200 Copenhagen N, Denmark.    Email: awalter@sund.ku.dk, manon.berns@sund.ku.dk

### Key points

- A mathematical model describing the role of synaptic protein tethers to localize transmitter-containing vesicles is developed based on geometrical considerations and structural information of synaptotagmin, Munc13 and SNARE proteins.
- Vesicle movement, along with tether association and dissociation, are modelled as stochastic processes, with tethers functioning independently of each other.
- Multiple tethers cooperate to recruit vesicles to the plasma membrane and keep them there: Munc13 and Syt as the longer tethers accelerate the movement towards the membrane, whereas short SNARE tethers stabilize them there.
- Model predictions for situations in which individual tethers are removed agree with the results from experimental studies upon gene knockout.
- Changing tether length or copy numbers affects vesicle trafficking and steady-state distributions.

## Introduction

The release of neurotransmitters from prfesynaptic terminals is essential for information transfer across chemical synapses (Abbott & Regehr, 2004). Before neurotransmitters can be released upon synaptic vesicle (SV) fusion, SVs need to be made release-ready by localizing towards the plasma membrane (PM) (docking) and functionally maturing (priming). These processes occur at specialized regions of the presynaptic terminal, termed active zones (AZs), and are regulated by different specialized proteins (Sudhof, 2013). How these proteins cooperate during SV docking, priming and fusion is still an open question.

The energy required for SV fusion is provided by the assembly of the neuronal SNARE complex. The complex consists of the v-SNARE synaptobrevin-2/VAMP2, which is attached to the SV via its transmembrane domain, and the t-SNAREs SNAP25 and syntaxin, which are associated with the PM (Jahn & Fasshauer, 2012; Sudhof, 2013). Besides its role in fusion, the N-terminal assembly of the SNARE complex is involved in SV docking and priming (Hernandez et al., 2012; Sørensen et al., 2006; Walter et al., 2010). SNARE complex formation is facilitated by Munc13

(Ma et al., 2011; Magdziarek et al., 2020; Wang et al., 2019), a large AZ protein mediating SV docking and priming (Aravamudan et al., 1999; Augustin et al., 1999; Bohme et al., 2016; Imig et al., 2014; Richmond et al., 1999; Siksou et al., 2009; Varoqueaux et al., 2002). Specifically, the role of Munc13 during docking and priming requires its MUN domain (Basu et al., 2005; Stevens & Sullivan, 2003). The activity of this domain is modulated by $Ca^{2+}$-calmodulin binding to the CaM domain of Munc13, diacylglycerol (DAG) binding to its $C_1$ domain and $Ca^{2+}$-phospholipid binding to its $C_2B$ domain (Junge et al., 2004; Rhee et al., 2002; Shin et al., 2010). Vesicle docking is also assumed to involve synaptotagmin 1/2 (Syt) (Chang et al., 2018; Chen et al., 2021; de Wit et al., 2009), which is well-known for its $Ca^{2+}$-dependent regulation of SV fusion (Geppert et al., 1994; Kobbersmed et al., 2022; Kochubey & Schneggenburger, 2011; Sudhof, 2013). The SNARE complex, Munc13 and Syt have all been suggested to form a bridge (tether) connecting the SV to the PM. The SNAREs form a protein tether upon (partial) assembly of the v- and t-SNAREs into one complex (Radecke et al., 2023). Munc13 is assumed to bridge two membranes by binding to the SV through its C2C domain (Padmanarayana et al., 2021; Quade et al., 2019) and the PM via its C1–C2B domains (Liu et al., 2016; Quade et al., 2019). Syt, which is attached to the SV via its transmembrane domain, can associate with the PM upon binding to $Ca^{2+}$ or the PM lipid phosphatidylinositol 4,5-biphosphate [$PI(4,5)P_2$] (Bai et al., 2004; Fernandez-Chacon et al., 2001; Li et al., 2006).

Because of differences in the protein structures, the SNARE, Munc13 and Syt tethers are associated with different lengths (Li et al., 2007; Quade et al., 2019; van den Bogaart et al., 2011). Correspondingly, tethers with different lengths linking the SV to the PM have been observed in electron microscopy (EM) images of several preparations (Cole et al., 2016; Fernández-Busnadiego et al., 2010; Fernández-Busnadiego et al., 2013; Gipson et al., 2017; Harlow et al., 2013; Papantoniou et al., 2023; Szule et al., 2012). Moreover, an increased number of tethers coincides with shorter SV–PM distances (Cole et al., 2016; Fernández-Busnadiego et al., 2013; Papantoniou et al., 2023; Szule et al., 2012), and removal of some of these proteins changes the distribution of SVs with respect to the PM (Bohme et al., 2016; Chen et al., 2021; Fernández-Busnadiego et al., 2013; Imig et al., 2014; Papantoniou et al., 2023; Siksou et al., 2009). Based on these observations, it has been proposed that, during docking, the SV becomes attached to the PM with a longer tether and that shorter tethers are formed sequentially, leading to shorter SV–PM distances (Fernández-Busnadiego et al., 2013; Papantoniou et al., 2023). The transition from longer to shorter tethers might resemble the transition from loosely docked to tightly docked states as proposed by Neher and Brose (2018). Thus far, this hypothesis has not been systematically investigated because of the limitations of currently available experimental techniques. To study SV docking, a resolution at the nanometre range is required, which only EM studies can provide. However, EM does not have the temporal information necessary to investigate how docking proceeds. To extend our insights beyond the scope of available experimental techniques, mathematical models integrating knowledge and insights originating from different types of experiments can be used.

In the present study, we used mathematical modelling to investigate the importance of tethers with different lengths for SV docking in resting synapses. We based the model on geometric considerations and structural information available on the assembled SNARE complex, Munc13 and $PI(4,5)P_2$-bound Syt. We allowed these proteins to form tethers at a rate depending on their lengths and the distance between the SV and PM. We assumed that tether formation constrained SV mobility, favouring SV–PM distances where tethers had optimal length. Even though our model disregards any putative interactions between the tethers, we found that engagement of more tethers shortened SV–PM distances, based on geometrical principles alone. Despite their independent engagement, simulation over time showed that the longer tether proteins in our model, Munc13 and Syt, typically engaged first with SVs approaching the PM. These longer tethers promoted the formation of the SNARE complex and increased the rates of SV trafficking towards the PM. Because all reactions in our model were implemented to occur stochastically and independently of each other, proteins could be removed from the model without breaking the continuity in it. Thereby, the model provides a unique way to compare model predictions to experimental data obtained upon removal of proteins. We show that our model predictions qualitatively align with SV–PM distributions following gene knockout (KO) or artificial extension of protein length in EM analysis or physiological measurements. Although these findings suggest that a major function in SV docking can be attributed to geometrical tether features alone, we show that Munc13 KO phenotypes are better matched when assuming functional interdependence with the SNARE tethers, highlighting a limitation of this simplified approach. Using a model incorporating realistic protein stoichiometry, we showed that stoichiometric changes affect SV trafficking and steady-state SV–PM distance distributions. In our model, these effects can be offset by adjusting the rates of tether formation and dissociation, meaning that these parameters cannot be determined independently. Despite its limitations, our model provides a useful starting point to investigate the relevance of protein tether cooperation in SV docking.

# Methods

## Model conceptualization

We aimed to investigate the importance of tethering filaments with various lengths in the trafficking of SVs to the PM in resting synapses. In the model, we included the SNARE complex (as a unit), Munc13 and Syt, which are all three proposed to tether the SV to the PM (Li et al., 2007; Quade et al., 2019; van den Bogaart et al., 2011). In our model, we propose that (I) the rate at which these tethers form depends on the distance between SV and PM and the length of the tether. Furthermore, (II) the tether lengths determine the localization of the SV from the PM if the tether filaments have formed. Below, we briefly describe how the functions describing these properties are derived and integrated into a model to simulate SV docking. A more detailed description of the model can be found in Model implementation.

## The effective height of Munc13, SNARE complex and Syt

The assembly rate of protein tethers, as well as the SV–PM distances at which a tether retains the SV, depend on the length of the protein segment responsible for tethering the SV to the PM. More specifically, these two functions are determined by how far the tether can reach from the PM, which we termed the *effective height* (illustrated in the inserts in Fig. 1*B*). For all proteins, we allowed the effective height to vary to indirectly take the flexibility of proteins into account.

The effective protein length of the SNARE complex depends on the extent of zippering between the v-SNAREs and t-SNAREs (Jahn & Fasshauer, 2012). Measurements indicate that the N-terminal parts of the individual SNARE motifs interact when two opposing membranes are <20 nm apart (Li et al., 2007; Wang et al., 2016), approximately corresponding to the sum of the lengths of the individual SNARE domains (Fig. 1*A*, left, 9 + 9 nm). However, stable SNARE complex formation is assumed to occur only when the two opposing membranes are <8 nm apart because, at this distance, a strong increase in the interaction force between two opposing, flat membranes with inserted v- and t-SNAREs was measured (Li et al., 2007). In the same experimental setting, the minimal distance between the two bilayers that could be reached was 2 nm (Li et al., 2007), corresponding to the thickness of the assembled SNARE complex (Fig. 1*A*, left). These data indicate that the SNARE complex, when at least partially but stably assembled, has an effective protein length of between 2 and 8 nm. How probabale lengths inbetween these extreme values are is unknown. We therefore simplified the situation by assuming a uniform probability density distribution between those values, that is, all effective heights between 2 and 8 nm existed with an equal probability (Fig. 1*B*, left). This simplification allows efficient mathematical handling but will ignore relevant biological features of the SNAREs, which might favour specific relative orientations.

Munc13 associates to the PM via its $C_1$ and $C_2B$ domains (Liu et al., 2016; Quade et al., 2019) and to the SV via its $C_2C$ domain (Padmanarayana et al., 2021; Quade et al., 2019). Based on the structure that is available from this segment [Protein Data Base (PDB) 717X] (Grushin et al., 2022)), we estimated a distance of ∼20 nm between the $C_1$−$C_2B$ and $C_2C$ domains (Fig. 1*A*, middle). Based on its structure and recent cryo-EM data, it has been proposed that in resting synapses Munc13 is typically oriented almost perpendicular to the PM (Camacho et al., 2021; Grushin et al., 2022; Xu et al., 2017). Molecular dynamics simulations have shown that the angle Munc13 has with the PM might fluctuate slightly (Camacho et al., 2021). This means that the distance the protein can span from the PM, the effective protein height, also slightly varies. To capture these properties in our model, we assumed the angle between Munc13 and the PM to follow a normal distribution with a mean of 80° and an SD of 5°. This distribution was used to calculate the probability density function (PDF) of the effective protein height shown in Fig. 1*B* (middle). Alternative conformational states were proposed to affect SV priming/docking (Camacho et al., 2021; Jusyte et al., 2023) but were not considered here, which is an additional simplification.

Syt1/2, which associates with the SV via its trans-membrane domain, can bridge the SV to PM in a $Ca^{2+}$-dependent manner and $Ca^{2+}$ independent manner (Araç et al., 2006; Honigmann et al., 2013; Lin et al., 2014; Nyenhuis et al., 2019; Seven et al., 2013; van den Bogaart et al., 2011). This latter occurs via the binding of its C2B domain to $PI(4,5)P_2$, which is located on the PM (Kobbersmed et al., 2022; Lin et al., 2014; van den Bogaart et al., 2011). Because we decided to focus this study on resting synapses, in which $Ca^{2+}$ concentrations are low and Syt probably does not bind $Ca^{2+}$ (van den Bogaart et al., 2012), we only included the $Ca^{2+}$ independent $PI(4,5)P_2$ bound state in our model. The exact effective protein length of $PI(4,5)P_2$-bound Syt is mostly affected by the conformational state of the linker domain. Some studies suggest that this linker must be folded to a large extent (Lin et al., 2014), whereas others suggest that it is mostly unstructured (Nyenhuis et al., 2019; Prasad & Zhou, 2020). Because of these uncertainties, we used a simplified approach and assumed the effective tether length for $PI(4,5)P_2$-bound Syt follows a uniform distribution between 5 nm (the thickness of the C2 domains; Fig. 1*A*, right) and 29 nm (the maximum length with a completely stretched linker domain; Fig. 1*A* and *B*, right). Again, although this

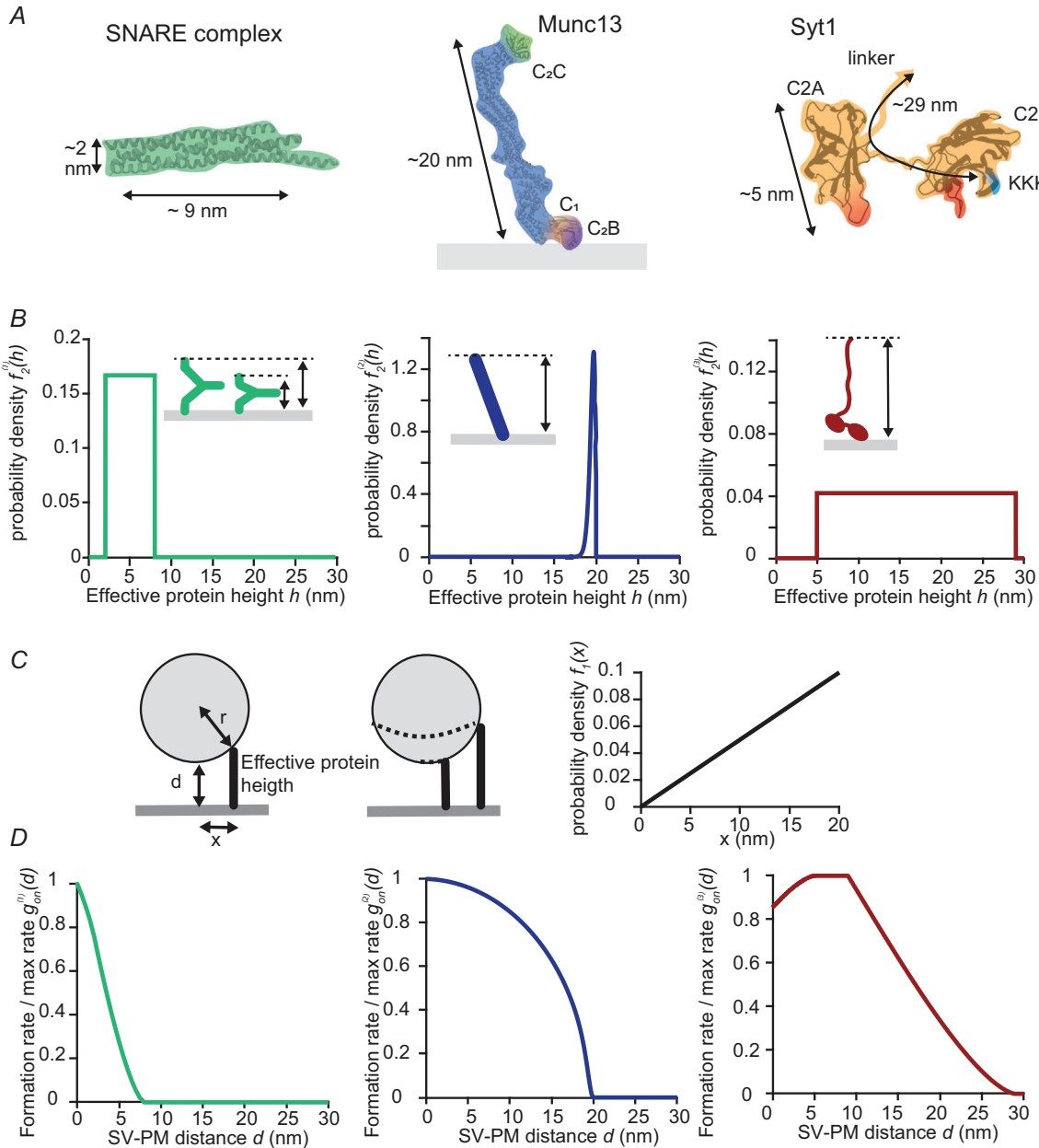

**Figure 1. The effective height of the SNARE complex, Munc13 and Syt determines the tether formation rate at different SV–PM distances**

*A*, protein structures of the assembled SNARE complex (PDB 1N7S, left), Munc13 (PDB 717X, middle) and Syt1 C2A and C2B domains (PDB 5CCG, right). The SNAREs tether the SV to the PM when at least partially assembled. Munc13 tethers the SV to PM through the simultaneous binding of its $C_2C$ domain to the SV and its $C_1$ and $C_2B$ domains to the PM. Via its polylysine motif (KKKK), Syt can bind to PI(4,5)$P_2$ on the PM. Syt is attached to the SV via a transmembrane domain connected to a long linker (not included in protein structure). *B*, probability density functions $f_2^{(i)}(h)$ of the effective heights of the SNAREs (left), Munc13 (middle) and Syt1 (right). The arrow in the inserts illustrates the effective protein height used here, which is defined as the distance between PM and the top of the protein. *C*, left: illustration of SVs tethered by a protein, here simplified to a stick standing perpendicular to the PM. The SV has a distance, *d*, from the PM, and the tether is formed at a distance *x* from the centre of the SV. *r* is the SV radius. The dotted lines in the right cartoons indicate the number of available binding locations for two tethers at different locations (*x*) from the projection of the SV centre to the plasma membrane. Right: probability density distribution $f_1(x)$ of *x*. *D*, normalized tether formation rate $g_{on}(d)$ of the SNARE complex (left), Munc13 (middle) and Syt (right) as a function of distance *d* between SV and PM. The formation rate is normalized to the maximum formation rate.

facilitates the simulation, relevant biological features of the protein favouring specific orientations might be ignored by this simplification.

### Derivation of distance-dependent assembling rates

Besides the length of the tether, the localization on the SV where tethers form affect whether a protein can bridge SV and PM at a given SV–PM distance. Assuming the SV is a perfect sphere with a radius of 20 nm (Takamori et al., 2006), the distance a protein needs to bridge is 20 nm larger when the tether forms on the outermost points of the SV compared to its centre. Because, in EM tethering, filaments are mainly observed on the lower SV hemisphere facing the PM (Harlow et al., 2013; Szule et al., 2012), we assumed that tethers only formed there. Additionally, we assumed that the probability of tether formation at a certain distance $x$ from the SV centre increases linearly with the corresponding area available for tether localization on the SV. This means that we assumed the probability of $x$ to be proportional to the circumference of a circle with radius $x$ (for the exact derivation of this function, see Model implementation, Fig. 1*C*). With this simplification, we ignored multiple aspects that could affect the localization of protein tethers on the SV, such as the preferred orientation of the domains inserting in the vesicular and plasma membrane (PM) and possible interactions between protein tethers.

By combining the PDFs of the effective protein lengths and the assembling location on the SV (see Model implementation), we could calculate a normalized assembling rate relative to the maximum assembling rate for each SV–PM distance for the different tethers (Fig. 1*D*). This means that given a specific SV-distance from the PM, the probability that a protein tether has the correct height to associate is considered.

### SV–PM distance distributions of an SV with different numbers and types of tethers

Once a tether forms, we propose that it will keep the SV within a specific range from the PM. This range depends on the effective protein height (Fig. 1*B*) and its association location on the SV (Fig. 1*C*). This means that localization of an SV with one tether formed can also be described by the functions shown in Fig. 1*D*. To estimate how probable it is that an SV tethered to the PM by multiple proteins is at a certain distance from the PM, we multiplied the probabilities of each individual protein tether to keep the SV at this SV–PM distance (see Model implementation for exact calculation). This means that, in the model, the tethers act as independent units. This simplification ignores possible tether interactions or spatial relationships between protein tethers. Besides the number and type of

assembled tethers, the localization of the SV with respect to the PM is highly influenced by membrane-repulsive forces (Leckband & Israelachvili, 2001). We included these forces in our model (see Model implementation), hindering the SV from approaching the PM closer than 2 nm (Fig. 2*A–C*).

### Combining the distance-dependent association rates and the SV–PM distributions to simulate SV tethering and trafficking

The trafficking and tethering of SVs is modelled as a reaction-diffusion jump process (Winkelmann & Schütte, 2020). Concretely, SVs move stochastically to and from the PM in steps of 0.5 nm. The probability of the SV moving towards the PM *vs.* from the PM is determined based on the SV–PM distribution corresponding to the number and type of tethers associated with the SV (example states shown in Fig. 2; for more details, see Model implementation). A random number between 0 and 1 determines, using these probabilities, whether the SV moves towards or from the PM, following the principles of the Gillespie algorithm (Gillespie, 2007). When no protein tethers were formed, SVs were allowed to move up to 100 nm away from the PM at a rate that is only affected by membrane repulsions close to the PM (Fig. 2*B*). Further distances were not considered. Note that the maximum SV–PM distance is an arbitrary setting in the model that will not influence the main properties of the model because most of the reactions will occur within 30 nm from the PM.

Besides stochastic distance-dependent assembly reactions (Fig. 1), we included the distance independent disassembly of protein tethers. The functions describing the tether assemblies as described above (see section on 'Derivation of distance-dependent assembly rates') are based on probabilities. We converted these probabilities to rates by multiplying the functions describing SV tethering to a fixed maximum tether assembly rate. The maximum rate of tether assembly and the rate of tether disassembly are unknown. We started by using the same assembly rates for all tethers. This simplification ignores that the interaction domains of the three tethering proteins are different, which is expected to result in markedly different rates. However, the main purpose of this simplified model was to investigate the general importance of different tether lengths. In the simulations shown in Figs 1*B* to 9*G*, the maximally achievable assembly rate was set to 40 s$^{-1}$ for all protein tethers (Table 1). In these simulations, the disassembly rate was set to 100 s$^{-1}$ for Munc13 and Syt tethers. We also adjusted these rate constants for the simulations shown in Fig. 9*H–M* in the version of the model including 35 SNARE tethers and seven Syt tethers (Table 1). The disassembly rate for the SNARE

complex was set to 50 s$^{-1}$ in the simulations shown in Figs 1*B* to 9*G*, aiming to account for the fact that, with the imposed model assumptions, the assembly rate of the SNARE complex will not reach its maximum rate at feasible SV–PM distances (>2 nm) (Fig. 1*D*, left, Fig. 2, and Table 1). A value of 76.7 s$^{-1}$ was chosen for the simulations shown in Fig. 9*H–M*. The values for maximal assembly rates and disassembly rates were selected based on initial model inspection and qualitative comparison of the model's predicted SV–PM distributions to the distributions obtained by Papantoniou et al. (2023). It is important to note that, for the purpose of steady-state SV docking distributions, the absolute values of the rates

are irrelevant, whereas the ratio between assembly and disassembly rates matters (indicated also by the parameter space exploration shown in Fig. 6). The model therefore does not provide reliable estimates on the exact time window in which these reactions happen.

In the model, we first included each nine copies of the SNAREs, Munc13 and Syt per release site/SV. This value was chosen based on estimates of the copy number of Munc13 per release site estimated with photoactivated localization microscopy (Sakamoto et al., 2018). We first decided not to include different copy numbers per protein to focus our study on the role of differences in the lengths of the involved protein tethers on SV trafficking

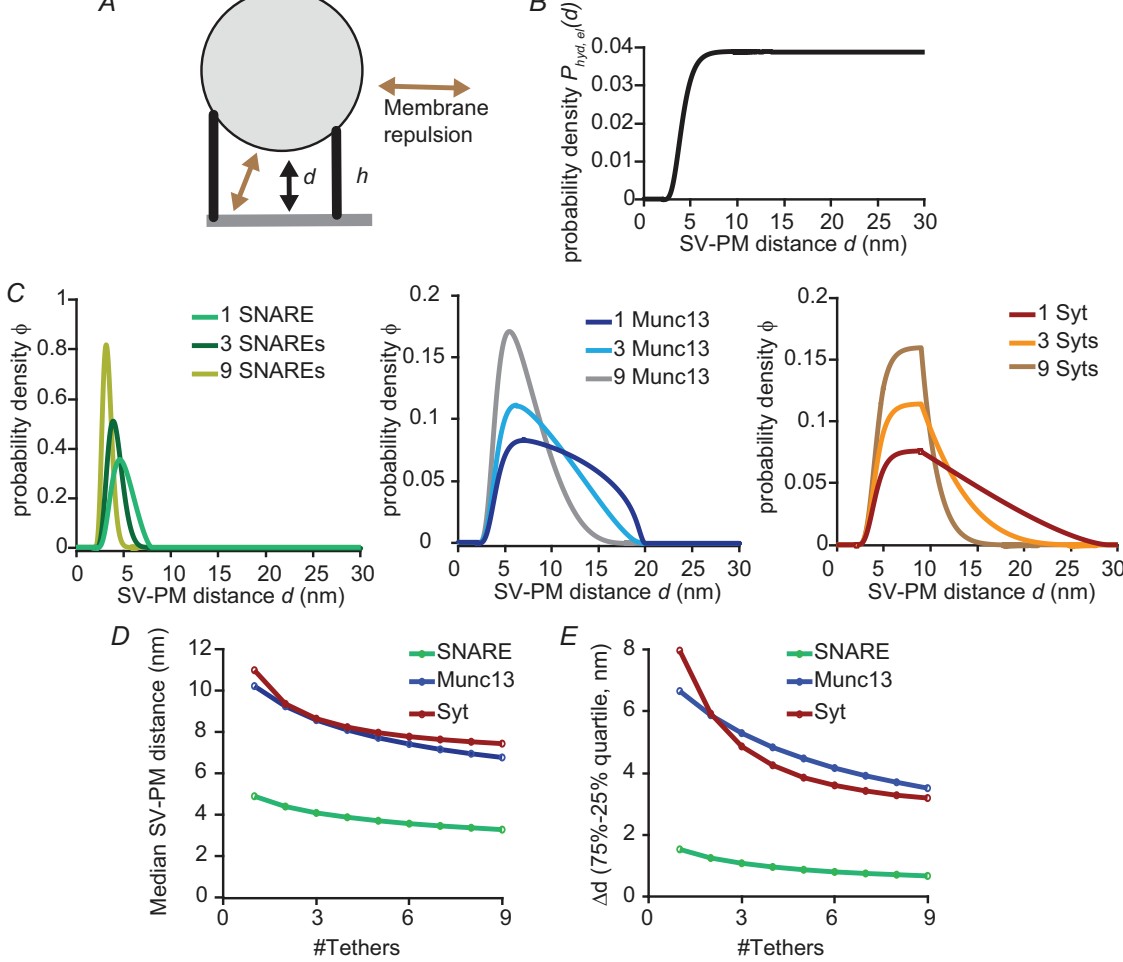

**Figure 2. Multiple tethers cooperate by promoting shorter SV–PM distances**
*A*, illustration of SV attached to the PM with two tethers with different effective protein heights $h$. The two proteins keep the SV at the same distance $d$. Membrane repulsion pushes the SV away when reaching closely to the PM. *B*, probability density distribution (PDF) of SV–PM distance $P_{hyd,el}(d)$ without any tethers formed. The resulting distribution is generated based on membrane repulsive forces (see Methods). *C*, PDFs $\phi(d, n^{(1)}, n^{(2)}, n^{(3)})$ of the SV–PM distances $d$ when one, three, or nine SNARE complexes (left: $n^{(1)} \in \{1, 3, 9\}$; $n^{(2)} = n^{(3)} = 0$), Munc13s (middle: $n^{(1)} = n^{(3)} = 0$; $n^{(2)} \in \{1, 3, 9\}$) or Syts (right: $n^{(1)} = n^{(2)} = 0$; $n^{(3)} \in \{1, 3, 9\}$) are tethering the SV to PM. *D*, median SV–PM distance of a vesicle with 1–9 SNAREs ($n^{(1)} \in \{0, \ldots, 9\}, n^{(2)} = n^{(3)} = 0$), Munc13 ($n^{(1)} = n^{(3)} = 0, n^{(2)} \in \{0, \ldots, 9\}$) and Syt ($n^{(1)} = n^{(2)} = 0, n^{(3)} \in \{0, \ldots, 9\}$) tethers. *E*, quantification of the SV–PM distribution width based on the distance spanned between the 25% and 75% quartile of the SV–PM PDFs.

**Table 1. Model parameters.**

| Parameter | Value | Reference | Notes |
|---|---|---|---|
| $n_{total}^{(1)}$ | 9 (35 in Fig. 9) | Takamori et al. (2006) for Fig. 9 | Copy number SNAREs |
| $n_{total}^{(2)}$ | 9 | Sakamoto et al. (2018) | Copy number Munc13 |
| $n_{total}^{(3)}$ | 9 (7 in Fig. 9) | Wilhelm et al. (2014) for Fig. 9 | Copy number Syt |
| $k_{on,max}^{(1)}$ | 40 (10.3 in Fig. 9*H–M*) s$^{-1}$ | * | Max SNARE assembly rate, adjusted parameter |
| $k_{on,max}^{(2)}$ | 40 s$^{-1}$ | * | Max Munc13 assembly rate, adjusted parameter |
| $k_{on,max}^{(3)}$ | 40 (51.4 in Fig. 9*H–M*) s$^{-1}$ | * | Max Syt assembly rate, adjusted parameter |
| $k_{off}^{(1)}$ | 50 (76.7 in Fig. 9*H–M*) s$^{-1}$ | * | SNARE Tether disassembly rate, adjusted parameter |
| $k_{off}^{(2)}$ | 100 s$^{-1}$ | * | Munc13 Tether disassembly rate, adjusted parameter |
| $k_{off}^{(3)}$ | 100 (88.6 in Fig. 9*H–M*) s$^{-1}$ | * | Syt Tether disassembly rate, adjusted parameter |
| Vesicular mobility | 16 000 s$^{-1}$ per 0.5 nm | Rothman et al. (2016) | |
| $P_0$ | 6e10 Pa | Mostafavi et al. (2017) | |
| Lambda | 0.2 nm | Mostafavi et al. (2017) | |
| $\lambda_d$, debye length | 0.8 nm | Mostafavi et al. (2017) | |
| $\varphi$, membrane potential | −60 mV | | |
| $T$ | 310 K | | Temperature |
| Length Munc13 | 20 nm | Grushin et al. (2022) | Estimated from PDB 7T7X |
| Angle Munc13 up (mean + std) | 80 ± 5° | | Estimated based on: 'Munc13 standing almost perpendicular' (Camacho et al., 2021) |
| Length Syt-PI(4,5)P$_2$ | 5–29 nm | van den Bogaart et al. (2011) | |
| Effective height $h$ SNARE complex | 2–8 nm (and 7–13 nm and 12–18 nm in Fig. 3) | Li et al. (2007) | |
| Radius SV r | 20 nm | Takamori et al. (2006) | |

*Adjusted model parameters.

toward the PM. In additional simulations shown in Fig. 9, we also included more realistic protein copy numbers of the vesicular proteins: 70 copies for the v-SNARE synaptobrevin-2 and 15 copies of Syt-1 of which we assumed half of those copies to be available for tether engagement with the PM (35 SNARE and 7 Syt tethers, respectively), because only one hemisphere of the SV will face the PM (Wilhelm et al., 2014; Takamori et al., 2006).

Stochastic simulations of the model were performed based on a Gillespie algorithm (Gillespie, 2007) with a single SV starting at 100 nm from the PM with no tethers formed. We simulated the model for 50 s and performed 100 repetitions for each setting (for more details, see Model implementation).

## Model implementation

**Estimation of effective protein lengths.** The relevant sizes of Munc13, SNAREs and Syts were estimated from PDB files 7T7X (Grushin et al., 2022), 1N7S (Ernst & Brunger, 2003) and 5CCG (Zhou et al., 2015), respectively, using the 3D viewer available via RCSB.org. For each protein, these values were used to determine the effective protein heights $h$ which are defined as the distance the proteins can span from PM.

For Munc13, $h$ was calculated from the length of the protein ($l_{protein}$) and an assumed angle the protein has with the PM, $\alpha$. We assumed that $\alpha$ followed a normal distribution with a mean 80 and a SD of 5° (angle), based on the suggestion that in low intracellular

$Ca^{2+}$-levels Munc13 stands close to perpendicular to the PM (Camacho et al., 2021). For each $\alpha$, height $h$ could be calculated using:

$$h(\alpha) = l_{\text{protein}} * \sin(\alpha) \tag{1}$$

where $l_{\text{protein}}$ corresponds to the length of the protein (20 nm). The probability of $h(\alpha)$ corresponds to the probability of $\alpha$.

The effective protein heights of Syt and the SNARE complex were implemented to follow a uniform distribution between the minimum and maximum length of the tethers: 5–29 nm for Syt-PI(4,5)P$_2$ (van den Bogaart et al., 2011) and 2–8 nm for the SNARE complex (Li et al., 2007).

**Derivation of distance-dependent assembly rates.** Using the estimated effective protein heights $h$ of the SNARE complex, Munc13 and Syt, we calculated the rate of tether formation by these proteins at different SV–PM distances. This rate was assumed to be proportional to the following term:

$$g_{\text{on}}^{(i)}(d) = \int f_1\left(\sqrt{r^2 - (r - h + d)^2}\right) * f_2^{(i)}(h)\ dh \tag{2}$$

where $i \in \{1, 2, 3\}$ is the index of the protein type denoting the SNARE ($i = 1$), Munc13 ($i = 2$) and Syt ($i = 1$) tethers, $f_2^{(i)}$ is the PDF of the effective protein height and $f_1$ is the PDF of the localization $x$ of the protein with respect to the SV centre, projected to the PM (Fig. 1C). The variable $d$ is the distance from the SV to the PM, measured at the vesicle's closest point to the PM. To derive this equation, we assumed that the SV is a perfect sphere with a fixed radius, $r = 20$ nm. We assumed that the probability of $x$ increases linearly with the number of assembling locations available at that value of $x$, which would correspond to the circumference of the circle with radius $x$. The PDF of $x$ can thus be written as:

$$f_1(x) = \frac{2x}{r^2} \tag{3}$$

For each of the protein tethers, the term $g_{\text{on}}^{(i)}(d)$ is normalized to the maximum value and subsequently multiplied by a constant representing the maximum assembly rate to obtain a function for distance-dependent assembly rates [see also eqn (10) below].

**Estimation of SV–PM distance PDFs for different number of tethers.** We next aimed to determine for all states an SV can be in (i.e. how many SNARE, Munc13 and Syt tethers have formed) what the probability is of observing the SV at different locations from the PM. To do so, we used the PDFs the effective protein heights and the assumption that the probability of an assembly location at $x$ distance from the SV centre increases linearly with the

number of assembly points, following Equation (3). Using those two considerations, we could calculate the PDF corresponding to the SV–PM distance when an individual tether has formed. This PDF, thus, can be calculated using eqn (2). We normalized this PDF to the area under its curve. To include multiple tethers, we used the simplified assumption that all tethers are bound independently of one another and that the probability of finding multiple such tethers of adequate height at a given SV–PM distance corresponds to the product of the PDFs.

The SV–PM distance is furthermore affected by membrane-repulsive energies. Important membrane repulsion forces include steric-hydration and electrostatic energies. The energy distribution corresponding to the steric hydration force can be calculated as (Leckband & Israelachvili, 2001):

$$E_{\text{hyd}}(d) = 2\pi\lambda^2 g P_0 e^{-d/\lambda} \tag{4}$$

In this equation, $g$ is a geometric factor, which, in our situation with a flat and a spherical membrane, corresponds to the radius of the SV. $P_0$ is the force amplitude and $\lambda$ is the decay length. The values of both parameters depend on the membrane composition (Rand & Parsegian, 1989). For our simulations, we set $\lambda$ to 0.2 nm and $P_0$ to 6e10 Pa, as used by Mostafavi et al. (2017). The electrostatic interaction between the membranes is calculated as (Leckband & Israelachvili, 2001):

$$E_{el}(d) = g Z_{\text{mb}} e^{-\frac{d}{\lambda_{\text{d}}}} \tag{5}$$

Here, $\lambda_{\text{d}}$ is the Debey screening length, which was set to 0.8 nm, corresponding to a salt concentration of 0.15 M (Leckband & Israelachvili, 2001). The $Z_{\text{mb}}$ factor is given by:

$$Z_{\text{mb}} = 9.38 * 10^{-11} \tanh^2(\varphi_0/107) \tag{6}$$

where $\varphi_0$ is corresponding to the membrane potential and was set to $-60$ mV. The membrane repulsion energies were converted into a PDF using the Boltzmann equation:

$$P_{\text{hyd,el}}(d) \sim e^{-\frac{E_{\text{hyd}}(d) + E_{\text{el}}(d)}{k_{\text{B}}T}} \tag{7}$$

where $k_{\text{B}}$ is the Boltzmann constant and $T$ the temperature (310 K). To calculate $P_{\text{hyd,el}}(d)$ exactly, the area under the function described in eqn (7) was normalized to 1 (in the range $d \in [0, 100]$nm). We assumed that the final PDF function of finding an SV at a given distance would correspond to the product of the PDF corresponding to the membrane repulsion energies with the PDFs of the protein tether lengths. This means that the final PDF

$$\phi(d, n^{(1)}, n^{(2)}, n^{(3)}) = P_{\text{hyd,el}}(d) \cdot \prod_{i=1}^{3} g_i^{n^{(i)}}(d) \tag{8}$$

is determined to the same degree by the membrane-repulsive forces and the effects of the protein tethers. Moreover, it also means that with this implementation the membrane-repulsive forces cannot be overcome by the addition of extra protein tethers. Because of this simplification, this model cannot be used to predict vesicle fusion, but only SV docking. Afterwards, the overall distribution was again normalized to the area under the curve.

**Simulation of SV movement and tether dynamics.** To simulate the process of SV docking, we combined the above-derived assembly rates and PDFs of SV localization in a time-homogenous Markov process. It was implemented following the Gillespie algorithm (Gillespie, 2007), allowing the stochastic movement of SVs, assembly and disassembly of tethers. The state of the system at time point $t$ is described by the distance $d$ the SV has from the PM, as well as the number of tethers that are formed by the SNARE complex, Munc13 and Syt. Each SV can undergo eight different reactions: (1) movement towards the PM, (2) movement away from the PM, (3–5) association with a SNARE, Munc13 or Syt tether and (6–8) disassembly of either of these tethers.

We implemented the movement of SVs in steps of 0.5 nm to and from the PM based on a rate of 16.000 s$^{-1}$, based on the estimated diffusion constants of SVs in hippocampal boutons (Rothman et al., 2016). For movements towards the PM, this rate was multiplied by:

$$m\left(d, n^{(1)}, n^{(2)}, n^{(3)}\right)$$
$$= \frac{\phi\left(d - 0.5, n^{(1)}, n^{(2)}, n^{(3)}\right)}{\phi\left(d - 0.5, n^{(1)}, n^{(2)}, n^{(3)}\right) + \phi\left(d, n^{(1)}, n^{(2)}, n^{(3)}\right)}$$

(9)

This ensures that SVs are more likely to move in the direction where the PDF [eqn (8)] has higher values. For movements away from the PM, the diffusion rate was multiplied by $1 - m$.

Given a distance $d$ of the SV to the PM, the rate of tether assembly was calculated by $k_{on}^{(i)}(d) \cdot (n_{total}^{(i)} - n(i))$ for

$$k_{on}^{(i)}(d) := g_{on}^{(i)}(d) \cdot k_{on, \, max}^{(i)}.$$

(10)

In this equation $g_{on}^{(i)}$ equals the assembly rate function for tether type $i$ [eqn (2)], $k_{on,max}^{(i)}$ is the maximal assembly rate of tether type $i$, $n_{total}^{(i)}$ is the copy number of protein $i$, and $n^{(i)}$ is the current number of tethers formed by that protein type. The tether disassembly rate was calculated with $k_{off}^{(i)} \cdot n^{(i)}$, independetly of $d$. To keep the focus of the model on the differences in effective protein heights, we first decided to set $k_{on,max}^{(i)}$, $n_{total}^{(i)}$ and $k_{off}^{(i)}$ equal for all included protein tethers, except for the SNARE

complex. For the SNARE complex, $k_{off}^{(i)}$ was reduced by half compared to the other proteins to correct for the fact that the SNARE complex cannot achieve its maximal assembly rate at SV–PM distances that can be reached by the SV (compare Figs 1 and 2). In these simulations, $n_{total}^{(i)}$ was set to 9 for all proteins included. This number corresponds to the estimated number of Munc13s available per release site (Sakamoto et al., 2018).

The copy numbers estimated for Syt and the vesicular SNARE protein synaptobrevin-2 are different to this and we therefore included additional simulations (Fig. 9) with $n_{total}^{(1)} = 35$ and $n_{total}^{(3)} = 7$, assuming that half of the total SV proteins found biochemically (70 synaptobrevin-2 and 15 Syt-1) (Takamori et al., 2006; Wilhelm et al., 2014) would be on the lower SV hemisphere and therefore available for tether formation.

We note also that the rate constants of tether assembly and disassembly are free parameters in the model. These parameters can balance changes in the total copy numbers in the sense that the long-term average number $\bar{n}^{(i)}(d)$ of attached tethers, given a fixed distance $d$, can be kept unchanged: As the tethers associate and dissociate independently of each other, this mean number of tethers is given by

$$\bar{n}^{(i)}(d) = n_{total}^{(i)} \cdot \frac{k_{on}^{(i)}(d)}{k_{off}^{(i)} + k_{on}^{(i)}(d)}$$

(11)

for each protein type, which results from the steady-state equation at distance $d$. Replacing the total number $n_{total}$ of available tethers by $\alpha \cdot n_{total}^{(i)}$ for a factor $\alpha > 0$, we can choose a new association rate $\frac{k_{on}^{(i)}(d)}{\alpha}$ and a new dissociation rate $k_{off}^{(i)} + (\alpha - 1)\frac{k_{off}^{(i)}}{\alpha}$ to keep the long-term average on the same value (note that this only works as long as $\alpha > \frac{k_{on}^{(i)}(d)}{k_{off}^{(i)} + k_{on}^{(i)}(d)}$ because, otherwise, we obtain a negative dissociation rate).

The rates of the eight possible reactions (spatial jumps upward and downward along with the binding and unbinding of the three protein types) were combined into a vector $b$. We denote the sum of all elements in $b$ by $b_0$. Using two random numbers $q_1, q_2 \in (0, 1)$ drawn from a uniform distribution using the built-in Matlab function *rand*, the time step to the next reaction and the type of the next reaction was determined. The time to the next reaction, $\tau$, is given by:

$$\tau = \frac{1}{b_0} \ln\left(\frac{1}{q_1}\right)$$

(12)

The index, $j$, of the reaction that is occurring is the first index that satisfies:

$$\sum_{j'=1}^{j} b_{j'} \geq q_2 b_0 \qquad (13)$$

According to the selected reaction, the state of the SV is updated, and $t$ is updated by $t + \tau$ Accordingly, $b$ and $b_0$ are updated and all other steps are repeated. This iterative process continues until $t > t_{\text{end}}$. By describing the diffusive movement of the SV as discrete spatial jumps, treated as reactions, the entire process can be characterized as a reaction-diffusion jump process; for further details, see (Winkelmann & Schütte, 2020).

**Simulation settings.** At the start of the simulation, we placed a single SV at a distance $d$ of 100 nm from the PM. At this location, the SV does not have any tethers formed. Using the above-described algorithm, we simulated the trajectory of this vesicle. For our simulations, we used a total time of 50 s. The simulated resolution of the SV–PM distance $d$ was 0.5 nm. In simulations trying to capture gene KO phenotypes, proteins were removed from the model by setting the copy number of these proteins to $n_{\text{total}}^{(i)} = 0$. For Munc13-, SNARE-simulations this holds both for Munc13 and the SNARE complex, $n_{\text{total}}^{(1)} = n_{\text{total}}^{(2)} = 0$. All simulations were performed in Matlab, version 2020b. For each setting, we used 100 repetitions of the simulation. Figures show the mean over these simulations and error bars indicate the standard error of the mean (SEM).

To run our model, we needed to adjust two main parameters: the maximal assembly and disassembly rates of the tethers. We varied the values of these parameters during a parameter sensitivity analysis (Fig. 6) and selected the set of parameter values matching the distribution of SVs in cryo-EM experiments in control, SNAP25 KO, and Munc13 double knockout (DKO) conditions (Papantoniou et al., 2023). Other model parameters were constrained based on experimental data and previous estimates (Table 1). To mimic high-pressure freezing and freeze-substituted (HPF-FS) EM, the same settings were used as in the original simulations. From these model simulations, we extracted for each time point the number of protein tethers assembled per SV (i.e. the SV state). Subsequently the PDF of the location of SV in the absence of membrane repulsive forces ($E_{\text{hyd}}(d) + E_{\text{el}}(d) = 0$) was determined for each of these SV states. All PDFs were multiplied by the proportion of time the SV spent in that specific state and subsequently summed together, to calculate the distribution of SV locations.

## Results

### Compared to the SNAREs and Munc13, Syt can form tethers at larger SV–PM distances

The SNARE complex, Munc13 and Syt, each have different distances they can span between the SV and the PM. From these three proteins, Syt has the largest possible effective height, and the SNARE complex has the smallest (Fig. 1$B$), implying that Syt tethers can form first when an SV approaches the PM (Fig. 1$D$). For the SNARE complex and Munc13, the tether formation rate increases the closer the SV gets to the PM. This is because smaller SV–PM distances allow the tethers to form further up on the periphery of the SV, where more binding locations are available (larger $x$ values) (Fig. 1$C$). The maximum binding rate of Syt to PI(4,5)P$_2$ is achieved when the SV is 5–10 nm away from the PM. This difference in the shape of the SNARE and Munc13 assembly functions originates from the assumption that the Syt-PI(4,5)P$_2$ tether has an effective height of up to 29 nm, which is larger than the SV radius ($r = 20$ nm). This means that, if an SV is closer than 9 nm from the PM, Syt is no longer able to form a tether in some of its extended orientations, leading to an overall decrease in probability.

### Multiple protein copies cooperate by reducing the SV–PM distance

We implemented the model such that formed tethers imposed preferred SV–PM distances based on the lengths of the associated tethers (Fig. 1$D$). Besides this, the preferred distance of an SV with respect to the PM, whether it is tethered or not, is affected by membrane-repulsive forces (Fig. 2$A$), which reduce the probability of an SV to be within 5 nm from the PM (Fig. 2$B$). Another aspect affecting the preferred distance is the number of tethers engaged. We allowed up to nine SNARE complexes, Munc13 and Syt tethers to form and assumed that all tethers were associated independently. This is a model simplification because these proteins are known to interact and be present in different stoichiometry (Sakamoto et al., 2018; Takamori et al., 2006), aspects which will be addressed below in additional simulations (Fig. 9). In general, more SV tethers favour shorter SV–PM distances in our model (Fig. 2$C$). When nine instead of one SNARE tethers were formed, the median SV–PM distance decreased from 4.89 to 3.27 nm (Fig. 2$C$). Similarly, for Munc13 and Syt, the median SV–PM distance decreased from 10.23 to 6.77 nm and 11 to 7.44 nm, respectively (Fig. 2$C$). Furthermore, we observed a smaller deviation in the SV–PM distance distributions when multiple tethers were assembled (Fig. 2$D$), indicating that SV locations become more confined to specific distances from the PM.

Taken together, these data show that in our model, protein copies can cooperate by reducing the SV–PM distance.

### Simulation of SV dynamics in a tether model

In the next step, we implemented a time-dependent version of the model based on Markov chain reactions to simulate SV movement and tether formation and dissociation. In these model simulations, SVs stochastically move to and from the PM (Fig. 3*Aa*). If SVs move in reach for tether interaction, tethers can assemble stochastically (Fig. 3*Ab*), following the functions shown in Fig. 1*D*. The assembly of protein tethers affects SV movement in favour of positions that are probable given the PDF $\varphi$ [see eqn (9)], which takes into account the optimal SV–PM distances given the associated tethers, as well as membrane repulsive forces. This can be seen *i* for example in Fig. 3*Ab* around timepoint 1.2 s, when the SNARE-tethers assemble, and the movement is confined to locations closer to the PM. After the binding of the first tether, the successive binding of additional tethers is typically observed (Fig. 3*Ab*). Occasionally, all tethers unbind (Fig. 3*A*, examples are indicated by the arrows).

Interestingly, parameter space exploration showed that only the ratio between assembly and disassembly rates is relevant for the distribution of SVs with respect to the PM (see below) (Fig. 6). This also implies that the time scale at which the SV moves and tethers form in the model is not constrained. In other words, the model only gives information on the predicted order of events and not their exact timing. The ratio between association and dissociation rate was determined by comparing the distance at which SVs spend the most time to the accumulation distance obtained with Cryo-EM in control, SNAP25KO and Munc13-1/2 DKO synaptosomes (see below) (Papantoniou et al., 2023).

### Increasing the tethering length of the SNAREs reduces the probability of short SV–PM distances

Experiments in which the distance between the transmembrane domain of the v-SNARE Syb2 was artificially enhanced by the addition of amino acid linkers showed a profound decrease in transmitter release (Deák et al., 2006; Kesavan et al., 2007). To investigate the consequences of artificial elongation of the SNARE-tethers in our model, we shifted the interval for the SNARE tether height *h* to longer distances (Fig. 3*B*). Increasing both the lower and upper bound of the height interval mimics the experimental conditions where the linker region between the Syb2 transmembrane domain and the SNARE binding motif was extended (Deák et al., 2006; Kesavan et al., 2007). When comparing the distribution of distances in our control model with versions in which the tethering length of the SNARES was extended by 5 or 10 nm (Fig. 3*C*), a rightward shift in the distribution of SV–PM distances to larger values was observed. Closer inspection revealed that the median distance was only mildly increased (Fig. 3*D*), but the probability of finding SVs within 5 nm of the PM markedly reduced with increasing tether lengths (Fig. 3*E*). By contrast, extending the length of the SNARE tethers neither affected the time it took SVs, placed at a 100 nm distance to the PM at the start of the simulation, to first reach a distance below 5 nm (Fig. 3*G*), nor did it affect the time it took before the first tethers formed (Fig. 3*H*). The decrease of SVs in close PM proximity qualitatively aligns with the observation that extending the linker region of Syb2 decreases the size of the readily releasable SV pool (Deák et al., 2006; Kesavan et al., 2007). However, as our model can only be used to simulate SV docking, not-fusion, additional effects linker extension may have on the efficacy of the fusion reaction cannot be captured. This might explain the relatively modest effects seen on the SV distribution compared to the dramatic loss of neurotransmitter release seen in the experiment.

### Tether removal in our model allows comparison with KO phenotypes

Because all reactions in our model were implemented to happen stochastically and independently from each other, the removal of individual proteins does not break the model's continuity. With that, our model allows investigating the role of different protein tethers during SV trafficking toward the PM by removing the proteins from our model. In the text below, we will refer to the model containing all proteins as the control model (where $n_{total}^{(i)} = 9 \; for \; i = 1, 2, 3$). We observed that removal of the SNARE complex (by setting $n_{total}^{(1)} = 0$) increased the proportion of larger SV–PM distances (Fig. 4*A*). At the selected parameter values, the most probable SV–PM distance increased from ∼6 nm under control conditions to ∼8 nm in the absence of the SNARE complex (Fig. 4*A*, left). Correspondingly, in the model simulations the median SV–PM distance also increased from 5 nm to 8 nm (Fig. 4*B*) and SVs spent less time within 10 nm from the PM upon removal of the SNARE complex (Fig. 4*C*). SNARE complex removal did not affect the time it took an SV to move from the start of the simulation at a 100 nm SV–PM distance to reach a PM distance of less than 5 nm, or until the first tether formed (Fig. 4*D* and *E*). By contrast, the average time an SV was tethered to the PM by at least one tether was reduced drastically upon SNARE removal (Fig. 4*F*). Together, these data indicate that the formation of the SNARE complex is crucial in bringing SVs close to the PM and stabilizing them in a tethered state.

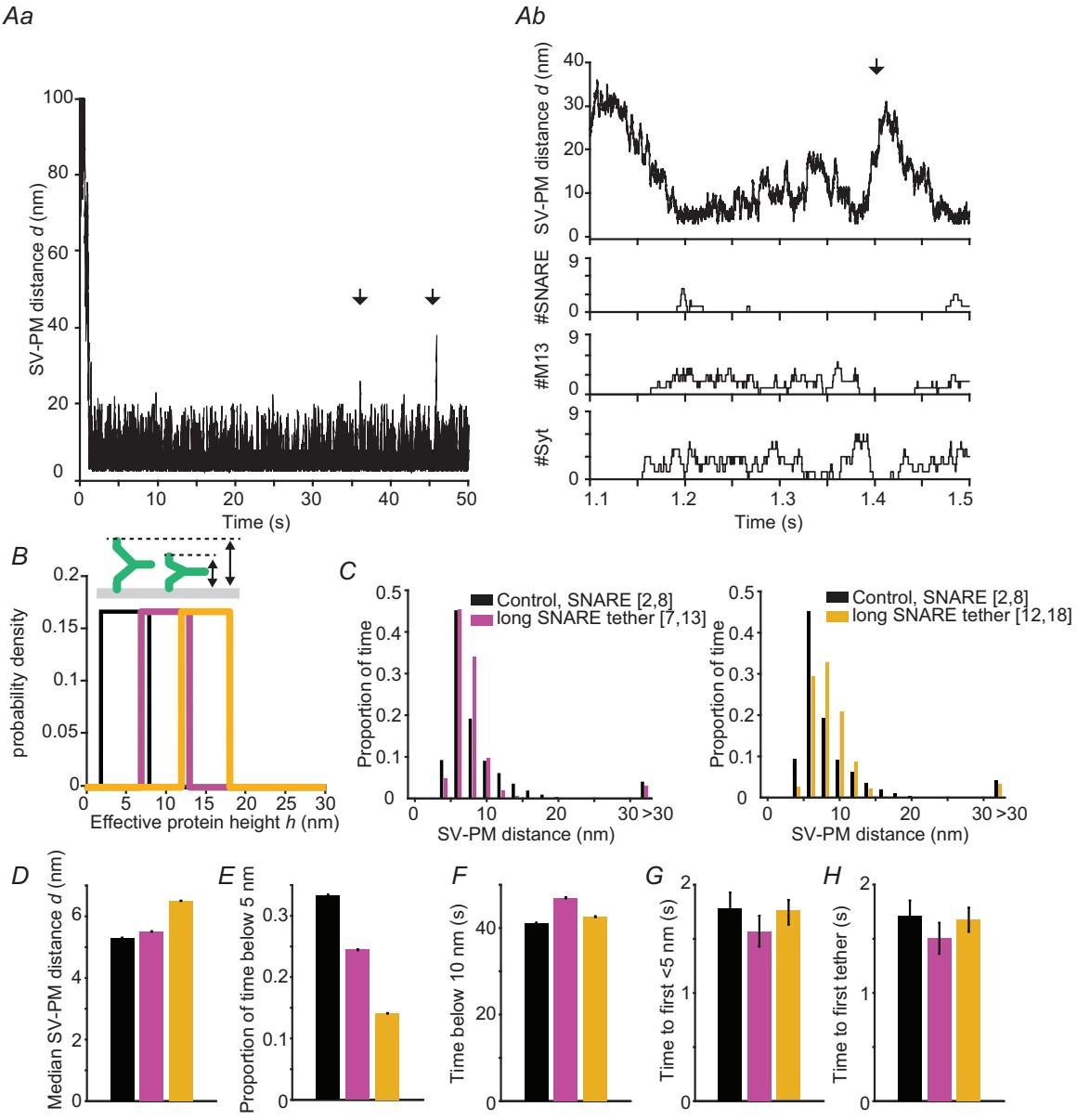

**Figure 3. Dynamic simulations of SV–PM distances for control simulations and simulations in which the tethering distance of the SNAREs were increased**

*Aa*, representative time-dependent localization of an SV (control) with respect to the PM for a model with up to nine SNARE, Munc13 and Syt tethers ($n_{total}^{(i)} = 9$ for $i = 1, 2, 3$). Arrows indicate examples of untethering events (i.e. unbinding of the last protein bridge). *Ab*, zoom-in of the simulation visualized in (*Aa*). Upper: the SV–PM distance during 0.4 s of the simulation. Lower: the number of SNARE complexes, Munc13 tethers (M13), and Syt tethers formed during this time frame. *B*, illustration of the investigated conditions in which the height of the SNARE linker was increased from the control condition (black, spanning a range of 2 to 6 nm) by 5 nm (red, spanning a range of 7 to 12 nm) and 10 nm (orange, spanning a range of 12 to 18 nm). *C*, proportion of time the SV spent at different SV–PM distances for control simulations and simulations in which the length of the SNARE tether was extended (interval [7,13] and [12,18] instead of [2,8] in the control setting. *D*, median SV–PM distance. *E*, the proportion of time the SV–PM distance was below 5 nm. *F*, the time the SV–PM distance was below 10 nm. *G*, time between the simulation onset and the first time the SV moves below 5 nm. *H*, time between the onset of the simulation and the formation of the first tether. Bars are calculated as the mean from 100 repetitions. Error bars depict the SEM.

In models without Munc13 ($n_{total}^{(2)} = 0$) or Syt tethers ($n_{total}^{(3)} = 0$), SVs spent less time close to the PM (Fig. 4*A* and *C*). Nonetheless, SVs within 30 nm from the PM were distributed similarly from the PM in these models compared to the control model, with the most observed distance being ∼6 nm (Fig. 4*A*). However, the median SV–PM distance was increased when either Munc13 or Syt were removed from the model (Fig. 4*B*). Just as in the model lacking the SNARE complex, in the models lacking Munc13 or Syt tethers, the time until the SVs moved below 5 nm or formed the first tether was similar to the control model (Fig. 4*D* and *E*). Additionally, removal of Munc13 or Syt from the model drastically reduced the average lifetime of tethered SVs (Fig. 4*F*). These data show that, although Munc13 and Syt removal did not cause a clear redistribution of SVs, both proteins were essential for increasing the time SVs were tethered. Moreover, both Munc13 and Syt appear to have very similar impact on the predicted SV–PM distances (Fig. 4), which can

be explained by the similar predictions of the SV–PM distributions when multiple Munc13 or Syt tethers have formed.

Besides a membrane tethering role, Munc13 also promotes SNARE complex assembly (Ma et al., 2011; Magdziarek et al., 2020; Wang et al., 2019). To explore the effect of both roles of Munc13 during SVs docking, we also simulated the model without Munc13 tethers and the ability to form SNARE complexes (Munc13-, SNARE-, i.e. removal of both the Munc13 tethers and the SNAREs tethers from the model, $n_{total}^{(1)} = n_{total}^{(2)} = 0$). The combined loss of Munc13 and SNARE tethering caused a redistribution of SVs towards larger SV–PM distances compared to control at the selected parameter settings, which agrees better with the EM analyses (Fig. 4*A*, left panel) (Imig et al., 2014; Papantoniou et al., 2023). We therefore conclude that, other than assumed in our simplified model of independent tether action, tethers functionally interact and this interaction is biologically relevant. We also note that, even under these conditions,

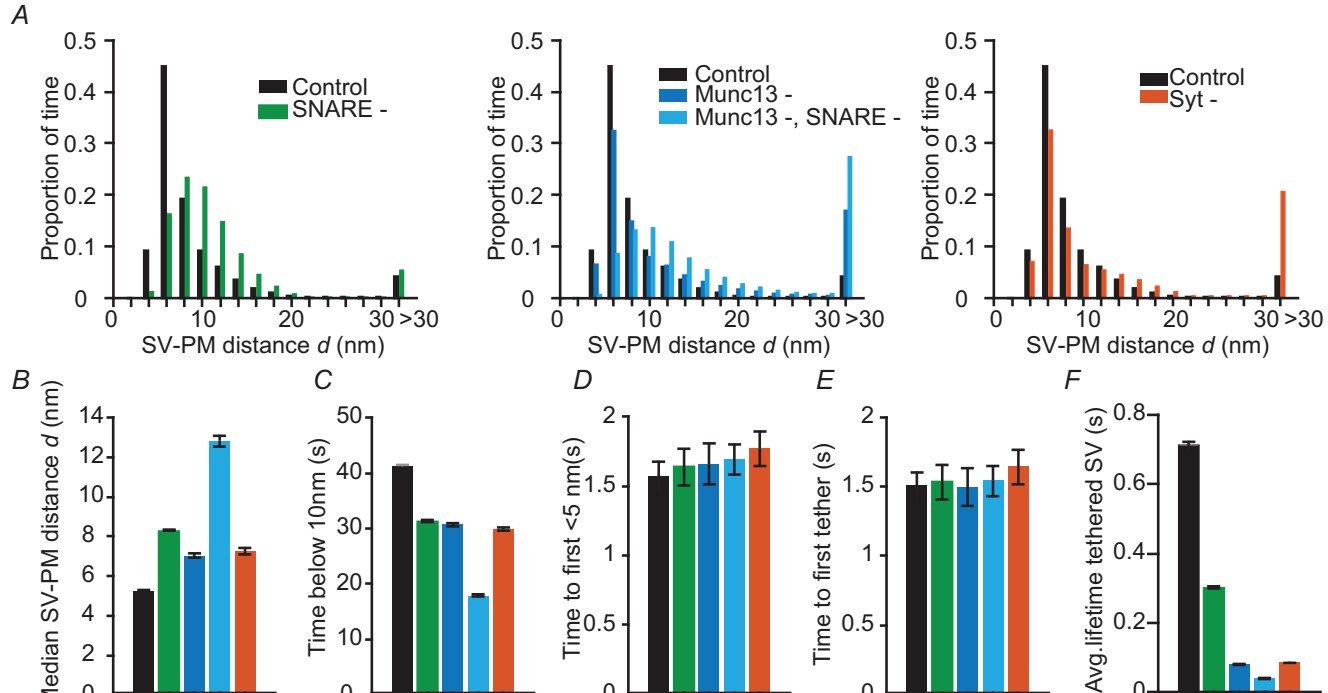

**Figure 4. Simulations of SV–PM distances for the control model and situations in which specific protein tethers were removed**

*A*, proportion of time the SV spent at different SV–PM distances for control simulations ($n_{total}^{(i)} = 9$, for $i = 1, 2, 3$) and simulations in which proteins were removed. Left: the control SV–PM distance distribution is compared to the SNARE complex removal condition ($n_{total}^{(1)} = 0$). Middle: control is displayed *vs.* two settings in which Munc13 is removed; one in which only the tethering function of Munc13 is lost (Munc13-, $n_{total}^{(2)} = 0$) and one in which both the tethering function and the SNARE complex assembly function are lost (Munc13-, SNARE-, $n_{total}^{(1)} = n_{total}^{(2)} = 0$). Right: the control SV–PM distribution is compared to the Syt removal condition ($n_{total}^{(3)} = 0$). *B*, median SV–PM distance. *C*, the amount of time the SV–PM distance was below 10 nm. *D*, time between the simulation onset and the first time the SV moves below 5 nm. *E*, time between the onset of the simulation and the formation of the first tether. *F*, average lifetime of a tethered SV. Model with up to nine tethers $n_{total}^{(i)} = 9$. Bars are calculated as the mean from 100 repetitions. Error bars depict the SEM.

the observed shift of SV–PM distances is still smaller than observed in Munc13DKO synaptosomes analysed with Cryo-EM (Papantoniou et al., 2023), which could be explained by long protein tethers such as RIM and bassoon which are not included in our model (see Discussion). Taken together, our simulation results imply that the SNARE complex is specifically required to accumulate SVs within 6 nm from the PM. In addition, tethers formed by the SNARE proteins, Munc13 and Syt are all required to increase the time SVs spend in a tethered state.

### Long tethers more probably form and promote the formation of other tethers

In our model, we allowed up to nine tethers to form per SV per protein (we also investigated other stoichiometries below, see Fig. 9). However, it was very rare that nine tethers of the same type were formed simultaneously (Fig. 5A). Interestingly, although all proteins in the model had the same tether association rate constant and copy numbers, fewer SNAREs were assembled compared to Munc13 and Syt tethers (Fig. 5B). This might be explained by the short range of SV–PM distances at which the

SNARE complex can form (Fig. 1D) and the strong membrane repulsions that exist in this range (Fig. 2A).

Removal of the SNARE complex from the model did not lead to dramatic changes in the number of Munc13 and Syt tethers (Fig. 5A). By contrast, the number of other tethers decreased in the absence of Munc13 and Syt (Fig. 5A). These data show that although the effects of Munc13 and Syt on SV–PM distributions in our model are minor, both proteins have a role in promoting the formation of other tethers. Under all conditions, Syt or Munc13 tethers most probably formed first (Fig. 5B), matching the hypothesis that longer protein tethers would have a higher probability of initial assembly. Note that these conclusions are sensitive to the tether formation dynamics in the model (stoichiometry, maximum assembly rate, disassembly rate), which we here decided to be similar for all proteins included.

### Parameter sensitivity and robustness of our interpretation

A particular difficulty of this model is that key parameters such as the rate constants for tether assembly and

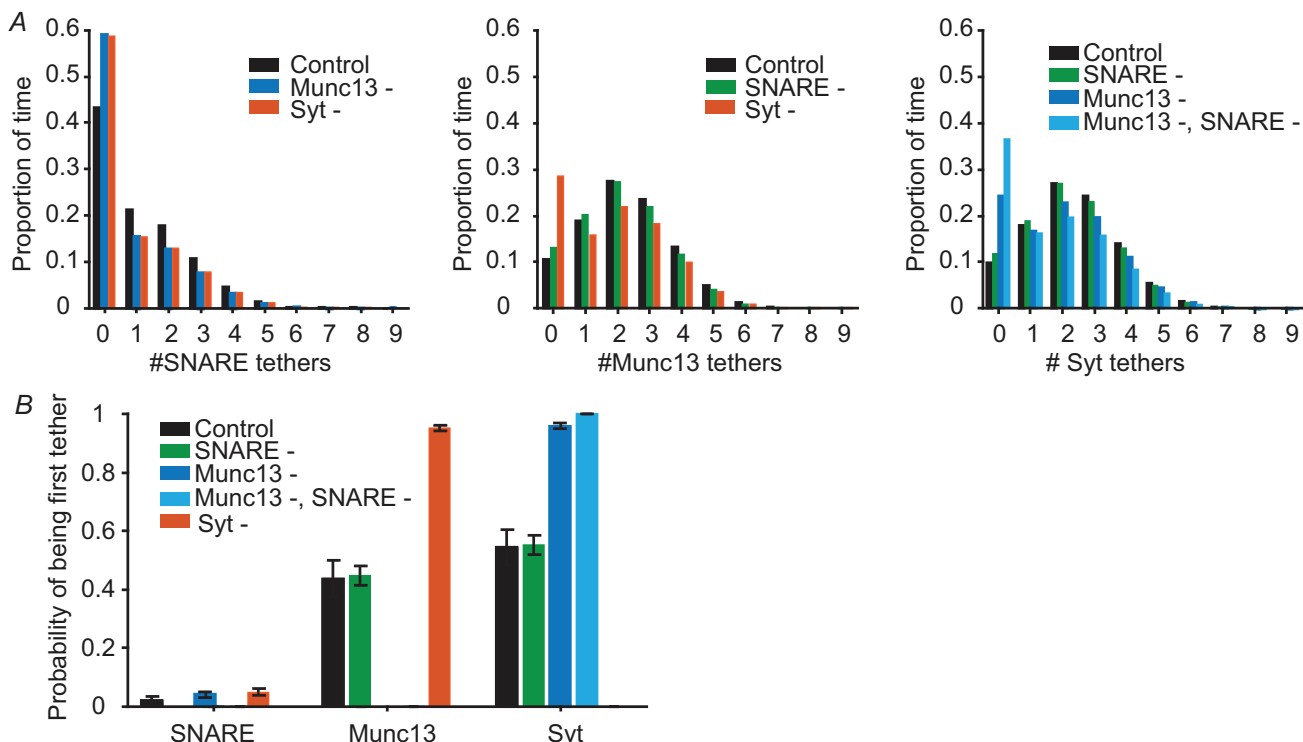

**Figure 5. Quantification of the number and type of tethers formed**

*A*, the proportion of time SVs have a certain amount of SNARE (left), Munc13 (middle) and Syt (right) tethers associated with it for control $n_{total}^{(i)} = 9$ for $i = 1, 2, 3$, and simulations following removal of the SNARE tethers (SNARE-, $n_{total}^{(1)} = 0$), Munc13 tethers (Munc13-, $n_{total}^{(2)} = 0$), Syt (Syt-, $n_{total}^{(3)} = 0$) or both the SNARE and Munc13 tethers (Munc13-, SNARE-, $n_{total}^{(1)} = n_{total}^{(2)} = 0$). *B*, the probability of the SNARE complex, Munc13 and Syt of being the first tethers assembling when the SV transits from an untethered state to a tethered state (occurs multiple times during a single simulation). Bars are calculated as the mean from 100 repetitions. Error bars depict the SEM.

disassembly are currently unknown. We, therefore, performed a sensitivity analysis by systematically varying these values and evaluating how their changes affect the predictions of the model (Fig. 6). We found that variation of the maximal tether association rate constant ($k_{\text{on,max}}^{(i)}$) or the dissociation rate constant ($k_{\text{off}}^{(i)}$) alone made the model very sensitive in its prediction of median SV–PM distances $d$ and the time SVs spent within 10 nm of the PM (Fig. 6*A* and *C*). On the other hand, the steady-state

model predictions were more robust against changes of the association rate constant when the dissociation rate was simultaneously changed to maintain a constant ratio. This shows that only the ratio of the rate constants affects the steady-state distribution of SV, which we compared to SV distributions seen in the synaptic ultrastructure (see Discussion). We also saw that, under the mutant conditions that we investigated here, the main observed effects such as the loss of membrane-proximal SVs upon

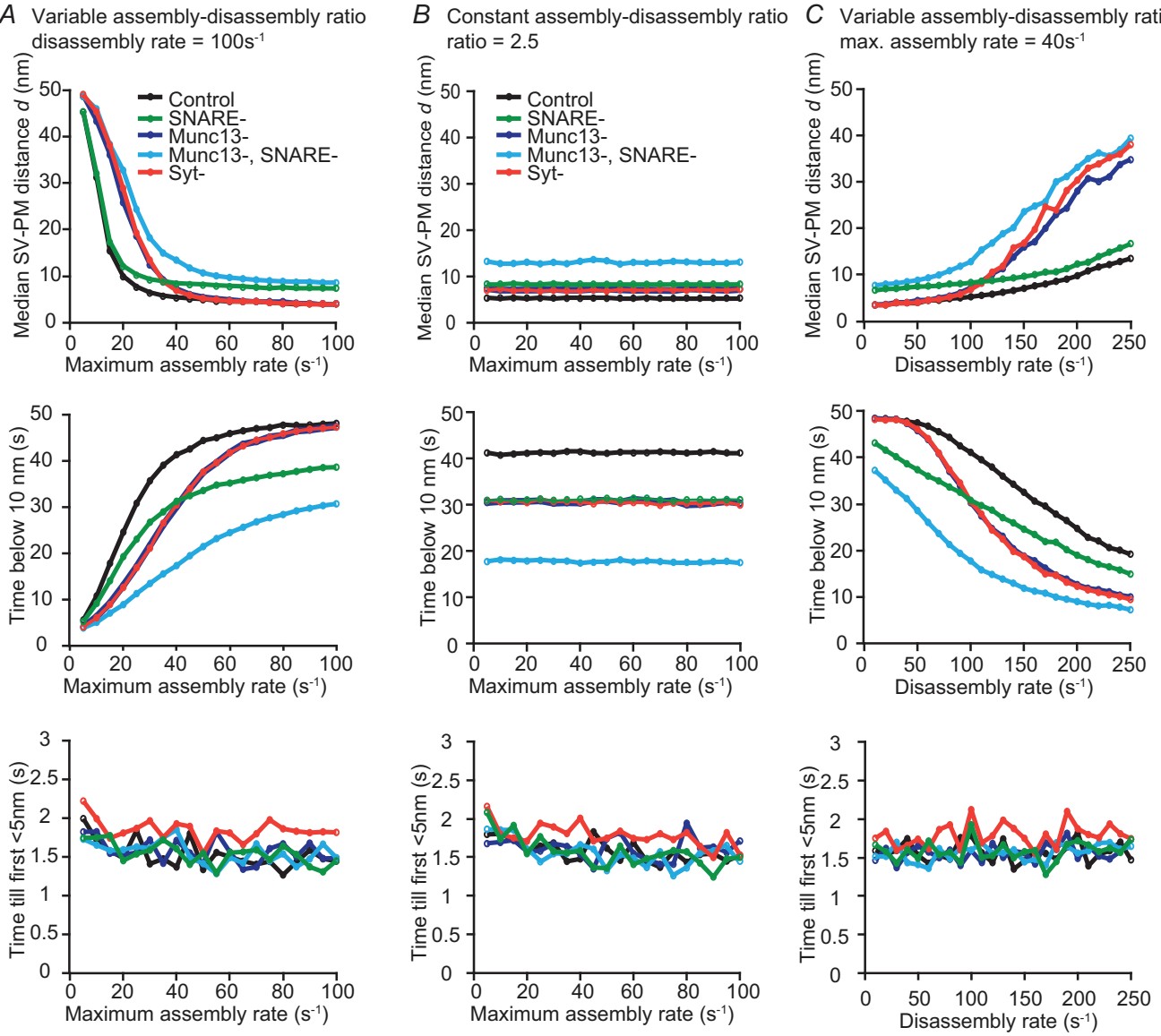

**Figure 6. Parameter sensitivity analysis**
*A*, effect of varying the maximal rate of tether assembly on the median SV–PM distance (top), time spent below 10 nm (middle) and time until the SV moves below 5 nm from simulation onset (bottom). In these simulations, the disassembly rate was set to 100s$^{-1}$ for Munc13 and Syt and to 50s$^{-1}$ for the SNAREs. *B*, same as for (*A*), but with varying the maximum rate of tether assembly together with the disassembly rate, aiming to ensure a constant same ratio between both values (ratio kept constant at 2.5 when compared to the kinetics of Munc13 and Syt). *C*, as for (*A*), but with varying the disassembly rate. The assembly rate was kept at 40 s$^{-1}$. Control: $n_{\text{total}}^{(i)} = 9$ for $i = 1, 2, 3$, SNARE-: $n_{\text{total}}^{(1)} = 0$, Munc13-: $n_{\text{total}}^{(2)} = 0$, Munc13-, SNARE-: $n_{\text{total}}^{(1)} = n_{\text{total}}^{(2)} = 0$, Syt-: $n_{\text{total}}^{(3)} = 0$. Points represent the average calculated over 100 repetitions.

removal of any of the tethers were robust across a broad range of tether association rates (Fig. 6*B*). We also saw that qualitative differences were preserved, such as the strongest defects were seen when assuming that Munc13 loss also affected SNARE tethering (Munc13-, SNARE-) (Figs 4 and 6). Although this demonstrates that this model can be useful for extracting steady-state features of SV docking, we again point out that the dynamic aspects suffer from the lack of knowledge of these rate constants. Therefore, the information we provide regarding timing needs to be taken with caution and rather interpreted regarding the sequence of reactions than their absolute timing.

## Model simulations in the absence of membrane repulsive forces predict shorter SV–PM distances

Much experimental data on SV–PM distribution is obtained from EM images obtained using HPF-FS samples. The dehydration that is required for this technique probably affects membrane-repulsive forces (Zuber & Lučić, 2019). To be able to compare our model simulations to these data sets, we decided to evaluate our model also in the absence of membrane-repulsive forces. From the previous simulations (Fig. 4), we extracted SV states (i.e. the number of tethers formed at each time point) and recalculated the localization of the SV using SV–PM distributions computed without membrane-repulsive forces. Under these conditions, SVs accumulated within 2 nm from the PM in control simulations (Fig. 7). A similar accumulation distance has been observed with HPF-FS EM (Chen et al., 2021; Imig et al., 2014; Siksou et al., 2009). Similar to the simulations analysed in the presence of membrane repulsive forces, simulations without Munc13 or Syt tethers did not affect the SV localization, but removal of the SNARE complex caused a loss of vesicles within 2 nm from the PM and an

accumulation of SVs at 6 nm (Fig. 7). The combined loss of Munc13 and SNARE tethers (Munc13-, SNARE-) caused the SV distributions to peak around 6–10 nm (Fig. 7). In conclusion, the absence of membrane repulsive forces does not change the qualitative effect that the removal of different proteins has on SV–PM distributions but leads to overall shorter SV–PM distances.

## The presence of long protein tethers accelerates SV trafficking to the PM

To obtain more insight into the exact role of the long tethers in our model (Munc13 and Syt) during SV movement towards the PM, we decided to simulate a version of the model only including SNAREs. To correct for the loss of a total number of protein copies, we simulated this SNARE-only model with 27 copies, besides the original nine copies. We compared these conditions to a model with no protein tethers ($n_{total}^{(i)} = 0$ for $i = 1, 2, 3$). When the model included only nine SNAREs no clear accumulation of SVs close to the PM was observed (Fig. 8*A*, left). This is quantified by an increased median SV–PM distance (Fig. 8*B*) and a strongly reduced time the SV spent below 10 nm compared to control simulations (Fig. 8*C*). Additionally, when the model only included nine copies of the SNARE t, SVs remained shorter in a tethered state compared to control simulations (Fig. 8*F*). These effects were complemented by a lower number of tethers formed in the model that includes nine copies of SNARE compared to control (Fig. 8*G*). When 27 copies of SNAREs were included in the simulations, the effects were reversed. The median SV–PM distance of proximal vesicles was slightly reduced compared to the control simulations (Fig. 8*B*), the lifetime of tethered vesicles was strongly increased (Fig. 8*F*) and the distribution over the number of formed tethers was close to one of the control model (Fig. 8*G*). A striking difference, however,

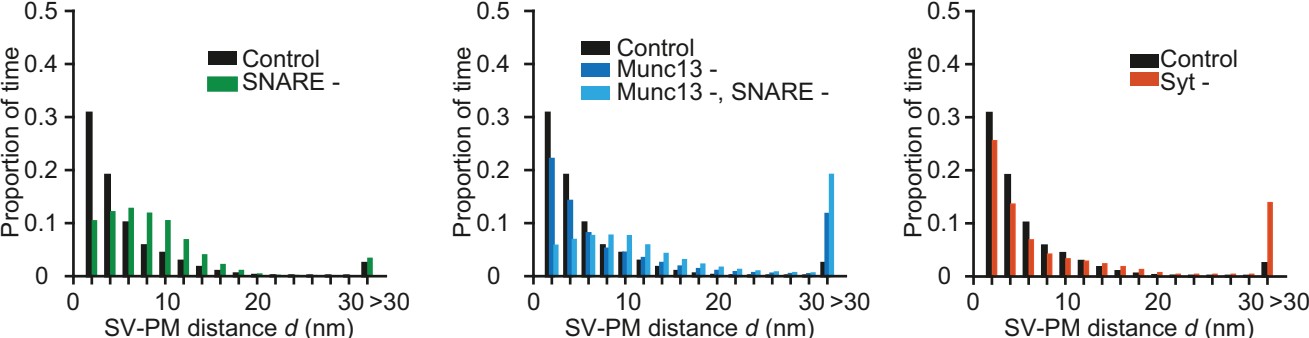

**Figure 7. SV–PM distributions when model is assessed without membrane repulsive forces**
SV–PM distance distributions for control model ($n_{total}^{(i)} = 9$ for $i = 1, 2, 3$) compared to simulations obtained without SNARE tethers (left, $n_{total}^{(1)} = 0$), without Munc13 tethers (middle, $n_{total}^{(2)} = 0$), without Munc13 and SNARE tethers (middle, $n_{total}^{(1)} = n_{total}^{(2)} = 0$) and without Syt tethers (right, $n_{total}^{(3)} = 0$). Membrane repulsion removed by setting $(E_{hyd}(d) + E_{el}(d) = 0)$. Bars represent the average of 100 repetitions.

between the control and simulations with only SNAREs is that it took longer until the SV moved below 5 nm for the first time after the onset of the simulation in the SNARE-only models (Fig. 8*D*). Importantly, this time was similar in SNARE-only models and a model without any protein tethers (Fig. 8*D*), indicating that SVs moved within this distance independently of the SNARE tethers. Similarly, the time it took until the first tether formed was increased in the SNARE-only simulation compared to control (Fig. 8*E*). Together, these results indicate that the longer protein tethers, such as Munc13 and Syt, are essential to accelerate the SV movement towards the PM.

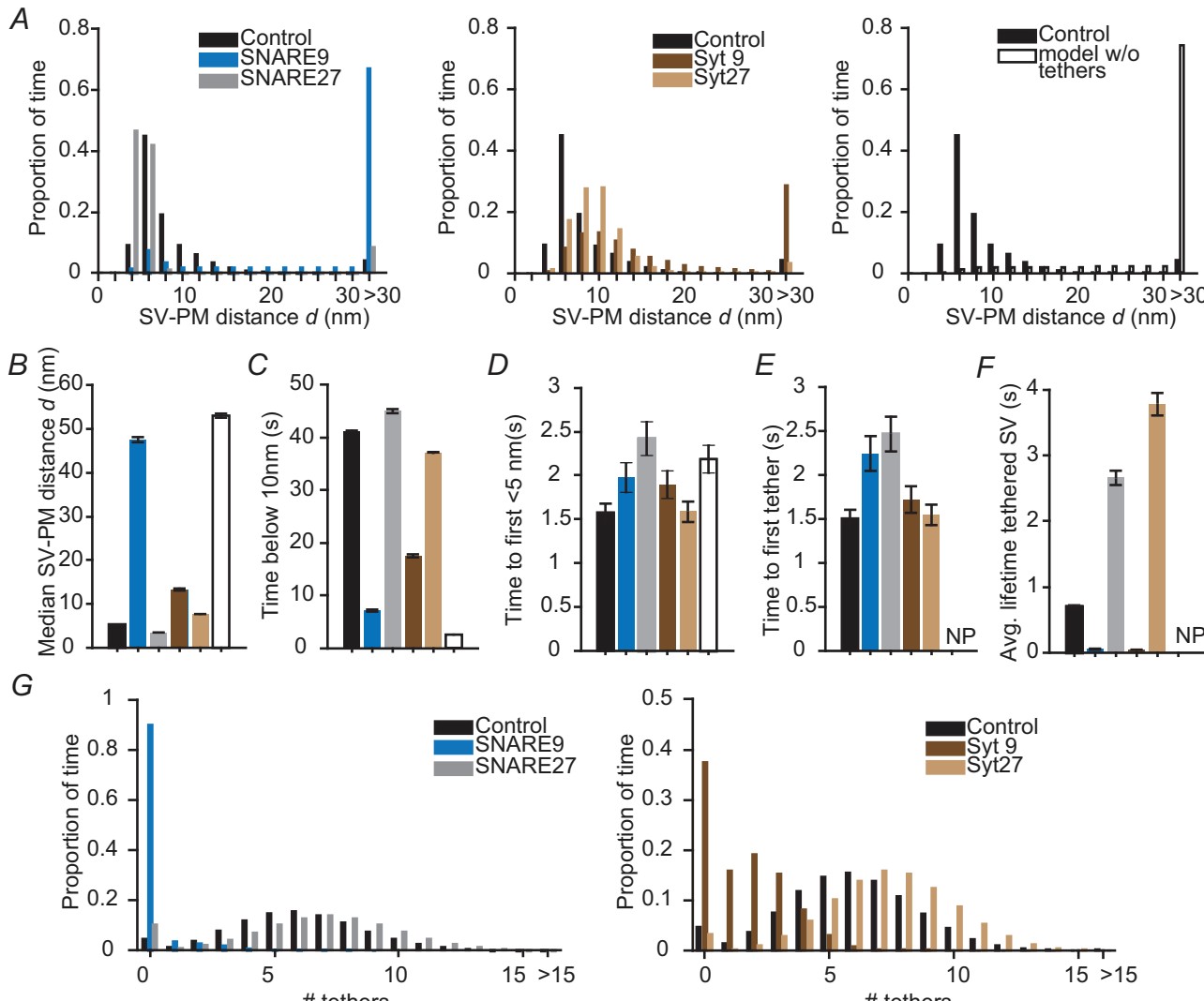

**Figure 8. Long tethers are required to increase docking rates, and short ones to reduce the SV–PM distance**

*A*, the proportion of time spent at different SV–PM distances for control simulations and simulations including only the same tether type. The control condition has up to nine tethers of all three types ($n_{total}^{(i)} = 9$ for $i = 1, 2, 3$). The blue and grey bars show results from models with up to 9 ($n_{total}^{(1)} = 9$) or 27 ($n_{total}^{(1)} = 27$) SNARE tethers in the absence of Munc13 and Syt tethers ($n_{total}^{(2)} = n_{total}^{(3)} = 0$). The dark and light brown bars show results from models with 9 ($n_{total}^{(3)} = 9$) or 27 ($n_{total}^{(3)} = 27$) Syts without any SNARE and Munc13 tethers ($n_{total}^{(1)} = n_{total}^{(2)} = 0$). The open bars show the results of the model simulated without any tethers ($n_{total}^{(i)} = 0$ for $i = 1, 2, 3$). *B*, median SV–PM distance *d*. *C*, amount of time the SV spent below 10 nm. *D*, the time between the start of the simulation and the first time the SV moved below 5 nm. *E*, time between the start of the simulation and the assembly of the first tether. *F*, the average time an SV was tethered to the PM with at least one tether. *G*, proportion of time the SV had a certain number of tethers for a model with only SNAREs (left) and only Syts (right). For control condition, the sum is taken of all tether types. Bars are calculated as the mean from 100 repetitions. Error bars depict the SEM. NP, not possible.

To investigate the importance of short protein tethers (the SNARE complex), we also simulated the model with only Syts. Compared to control simulations, models with only Syts showed a shift toward larger SV–PM distances (Fig. 8*A* and *B*). By contrast to the SNARE-only simulations, models that only included Syts did not show indications of delayed movement of SVs within 5 nm of the PM (Fig. 8*D* and *E*). Overall, these results indicate that specifically long protein tethers are required to increase the rate of SV trafficking towards the PM and the short ones (e.g. the SNARE complex) to bring SVs close to the PM.

### En route to a biologically more realistic model with different protein stoichiometries

Up to this point, we have assumed an identical total copy number of proteins for all three tether types under consideration, $n_{total}^{(i)} = 9 \; for \; i = 1, 2, 3$. This simplification probably does not reflect biological reality. We therefore adapted the model to include more realistic protein stoichiometries for the tethering proteins. In addition to the nine Munc13 proteins reported by (Sakamoto et al., 2018), we considered 70 copies of Syb2 and 15 copies of Syt1 per SV based on biochemical data (Takamori et al., 2006; Wilhelm et al., 2014). The relevant number of SNARE and Syt proteins available for SV–PM tethering in the model was set to $n_{total}^{(1)} = 35$ and $n_{total}^{(3)} = 7$, reasoning that 50% of these proteins would be on the lower SV hemisphere and able to interact with the PM. In trajectories generated by this version of the model, we observed stable attachment of SVs close to the PM and the formation of more SNARE tethers (Fig. 9*A*). Compared with the control model with up to nine tethers each, the distribution of SV–PM distances was shifted towards lower values (Fig. 9*B*) and the median SV–PM distance decreased (Fig. 9*C*). The proportion of time SVs spent close to the PM was increased with the effect being strongest within the first 5 nm (Fig. 9*D* and *E*). There was a slight tendency for delayed SV trafficking from the start of the simulations until SVs were within 5 nm of the PM and for the formation of the first tether (Fig. 9*F* and *G*). Thus, our results demonstrate that changing the number of available protein tethers alters the dynamics and steady-state distribution of SVs.

Considering that the rate constants of tether assembly and disassembly are free parameters in the model, we used the reweighting expressions introduced in the Methods section to recalculate these rate constants to achieve the same average number of associated tethers $\bar{n}^{(i)}$ as for the version of the control model where the maximal number of tethers was 9 each. The result of the simulations following this scaling of rate constants is shown in Fig. 9*H–M* in comparison with our simplified model with

each up to nine tethers. Overall, we see very similar steady-state behaviour for both versions of the model (see especially Fig. 9*H*). We note that, to achieve similar agreement with more SNARE proteins, the maximal rate constant of SNARE tether assembly was proportionally reduced, and the rate of SNARE tether disassembly was slightly increased (Table 1). By contrast, with fewer Syt tethers, their maximal assembly rate constant was proportionally increased and dissociation rate constant slightly reduced the dissociation rate constant (Table 1). We thus show that our simplified model with up to nine tethers each can capture situations with heterogenous tether numbers by adjusting the tether association and dissociation rate constants and that the conclusion drawn from this simple model can, therefore, be extended to biologically more plausible situations.

## Discussion

### Docking typically occurs through the sequential assembly of tethers with different lengths

SV docking to the PM is a complex process during which the SV localizes towards the PM. Based on EM data, it has been hypothesized that SV docking involves the sequential formation of longer to shorter tethers (Fernández-Busnadiego et al., 2013; Papantoniou et al., 2023). The systematic investigation of this hypothesis has been hampered by the lack of temporal information in EM. To study the dynamics of SV trafficking to the PM, we generated a mathematical model describing SV movement to and from the PM based on the formation of tethers with different lengths. In our model, we included the (partially) zippered SNARE complex, Munc13, tethering the SV to the PM via simultaneous binding of its $C_1/C_2B$ domain to the PM and its $C_2C$ domain to the SV, and Syt, bridging the two membranes in a $Ca^{2+}$ independent manner by binding to $PI(4,5)P_2$. We conclude that the longer protein tethers in our model, Munc13 and Syt, most probably form the first tethers connecting SVs to the PM. This state might correspond to an intermediate state of SV priming/docking (e.g. the loosely docked state introduced by Neher and Brose, 2018). The formation of these long tethers enhanced SV trafficking toward the PM and promoted SNARE tethering which in turn was essential to stabilize SVs at the PM. SNARE interaction also coincides with SV priming (Sørensen et al., 2006; Walter et al., 2010) and therefore the sequential assembly of longer to shorter tethers we see as a typical route for SVs to approach the PM aligns well with sequential models of the SV release process (Kobbersmed et al., 2020; Miki et al., 2018; Walter et al., 2013). A distinguishing feature of our model here is that our tethers were implemented to engage independently and stochastically and not in a fixed sequence. As such, the route that SVs typically take

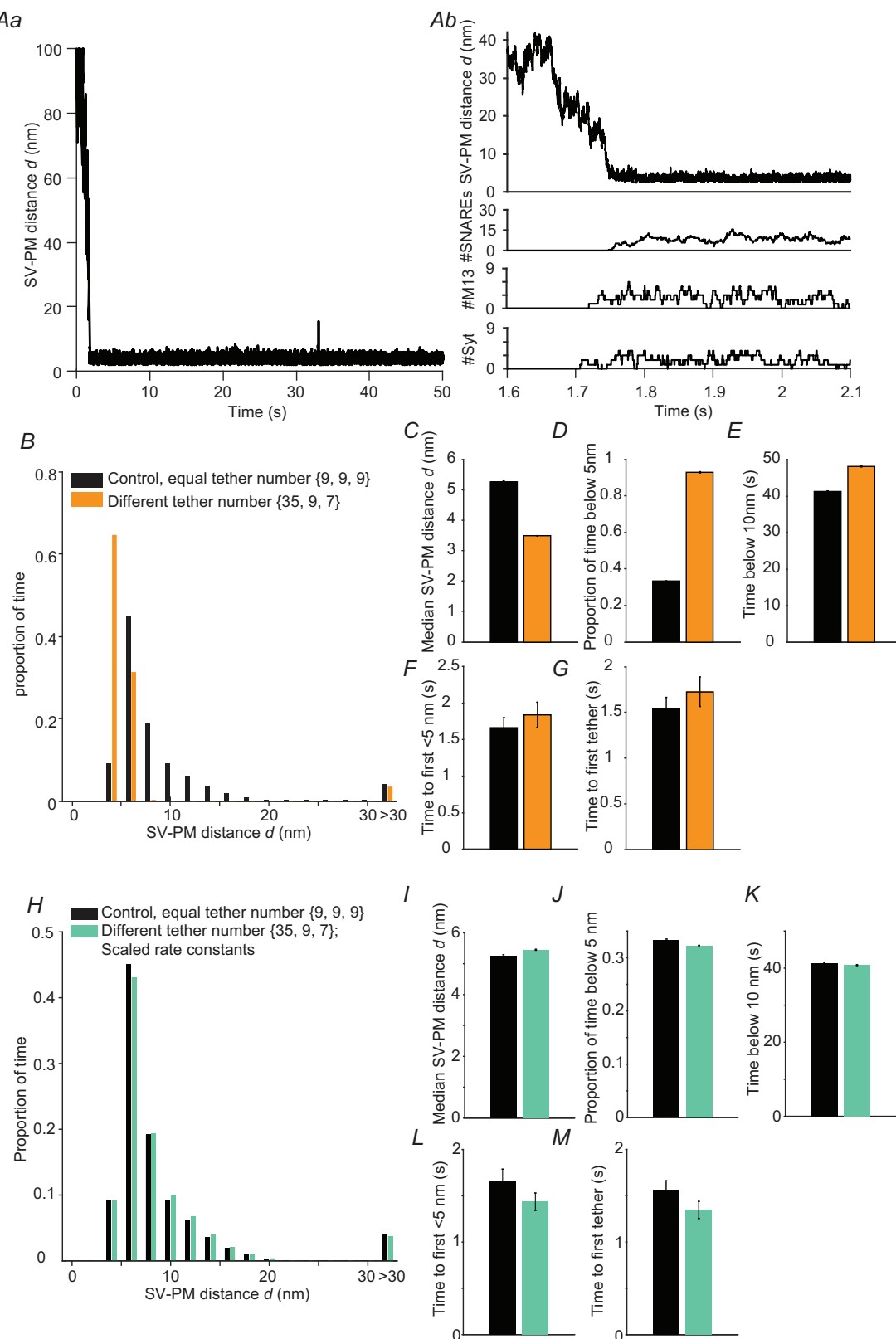

**Figure 9. A model with up to 35 SNARE tethers, nine Munc13 tethers and seven Syt tethers**

*Aa*, representative time-dependent localization of an SV with respect to the PM in a model where the maximal number of tethers is 35 for the SNARE tethers $n_{total}^{(1)} = 35$, 9 for Munc13 tethers $n_{total}^{(2)} = 9$ and 7 for Syt tethers

$n_{\text{total}}^{(3)} = 7$. *Ab*, zoom-in of the simulation visualized in (*Aa*). Upper: the SV–PM distance during 0.4 s of the simulation. Lower: the number of SNARE complexes, Munc13 tethers (M13) and Syt tethers formed during this time frame. *B*, proportion of time the SV spent at different SV–PM distances for control simulations where up to nine SNARE, Munc13 and Syt tethers can form (black, $n_{\text{total}}^{(i)} = 9$ for $i = 1, 2, 3$) and for simulations where up to 35 SNARE tethers, 9 Munc13 tethers and 7 Syt tethers can form (orange, $n_{\text{total}}^{(1)} = 35, n_{\text{total}}^{(2)} = 9, n_{\text{total}}^{(3)} = 7$). *C*,, median SV–PM distance. *D*, the proportion of time the SV–PM distance was below 5 nm. *E*, the amount of time the SV–PM distance was below 10 nm. *F*, time between the simulation onset and the first time the SV moves below 5 nm. *G*, time between the onset of the simulation and the formation of the first tether. *H–M*, same as (*B–G*) but with adjusted rate constants $k_{\text{on,max}}^{(1)}, k_{\text{on,max}}^{(3)}, k_{\text{off}}^{(1)}$ and $k_{\text{off}}^{(3)}$, to achieve similar behaviour as the control model with up to nine tethers each. For parameter values, see Table 1. Bars are calculated as the mean from 100 repetitions. Error bars depict the SEM.

to move towards the PM is not predetermined but a mere consequence of different tether lengths. Thus, although the average trajectory of the SV in preparation for neurotransmitter release might very well feature defined states through which SVs typically propagate sequentially, our model opens the possibility that these are not predetermined, and alternative routes exist. Sequential pool models predict that the relative population of different docked/primed SV states determines synaptic output and short-term plasticity (Kobbersmed et al., 2020; Lin et al., 2022; Neher, 2024; Pulido & Marty, 2018) and our results here show that this changes when either the number or length of tethers is altered. Consistent with this, changes in Munc13 conformation, local abundance or density coincide with presynaptic potentiation and altered short-term plasticity (Böhme et al., 2019; Camacho et al., 2021; Dannhäuser et al., 2022; Jusyte et al., 2023; Fukaya et al., 2023), opening the interesting possibility that some forms of presynaptic plasticity rely on tether modulation.

## Model predictions align with experimentally obtained SV–PM distributions

Most of our current understanding of SV docking can be attributed to research using genetically modified model organisms lacking specific protein components. Ideally, a mathematical model describing SV docking can replicate these experimental conditions. Previous models are based on predefined reaction chains (Hallermann et al., 2010; Pan & Zucker, 2009; Pulido & Marty, 2018; Walter et al., 2013), which break when individual components are removed. Consequently, to mimic genetic protein ablation, additional assumptions about the functional impact of protein removal are required. Because the model presented here is not based on a predefined reaction scheme, we could directly remove proteins from it to mimic experimental conditions.

In this work, we removed the SNAREs, Munc13 and Syt from our model. We constrained the unknown parameter values in our model based on its correspondence to cryo-EM data (Papantoniou et al., 2023). Additionally, we evaluated our model in the absence of membrane-repulsive forces, to

compare the results to HPF/FS EM. In the presence of membrane-repulsive forces, SVs accumulated at ∼6 nm in control setting and ∼8 nm upon removal of the SNARE complex, corresponding quantitatively to previous cryo-ET results (Papantoniou et al., 2023). Moreover, the predicted localization of SVs in the absence of repulsive forces in both control and SNARE removal simulations (accumulation at <2 *vs.* ∼6 nm) also corresponds with the HPF/FS results (Chen et al., 2021; Imig et al., 2014). Removal of Syt tethers from our model did not cause a redistribution of SVs with respect to the PM neither with, nor without membrane-repulsive forces. This aligns with observations in Syt KO where no clear change in the relative distribution of SVs close to the PM was observed by EM analysis (Chang et al., 2018; Chen et al., 2021; Imig et al., 2014). However, in these experiments, a loss of vesicles close to the PM is detected (Chang et al., 2018; Chen et al., 2021; Imig et al., 2014), which were suggested to be caused by the increased spontaneous release consuming SV close to the PM in SytKO synapses (Imig et al., 2014). In our model simulations, Munc13 removal only affected the SV localization when assuming that this also disrupted SNARE complex assembly such that both Munc13 and SNARE tethering were lost (Munc13-, SNARE-). In these simulations, SVs accumulated at ∼10 nm in the presence of membrane-repulsive forces and at ∼6–10 nm in the absence of those. Similar, although larger, redistributions of SV locations have been recorded in Munc13DKO synapses with cryo-EM and HPF/FS (Imig et al., 2014; Papantoniou et al., 2023; Siksou et al., 2009). The differences between the experiment and simulations are probably a result of the simplifications in our model, but the qualitative correspondence indicates that the roles of Munc13 in tethering and SNARE complex formation are essential for SV trafficking. Previous experiments indeed confirm that the role of Munc13 extends beyond a membrane tethering function (Magdziarek et al., 2020). Taken together, our model can reproduce SV localizations in control synapses and those lacking the SNAREs, Munc13 and Syt obtained with Cryo-EM and HPF-FS-EM. This correspondence with experimental data provides validation of our model to study the role of the SNARE complex, Munc13 and Syt during SV docking.

## Multiple protein copies cooperate by reducing the SV–PM distance

Multiple copies of the SNAREs, Munc13 and Syt are available per SV either on the SV itself or on the corresponding release site (Sakamoto et al., 2018; Takamori et al., 2006; Wilhelm et al., 2014). Evidence strongly indicates that multiple of these copies need to cooperate to regulate SV docking and fusion (Bouazza-Arostegui et al., 2022; Kobbersmed et al., 2022; Li et al., 2021; Mohrmann et al., 2010; Wu et al., 2017). In the first implementation of our model (Figs 3–8), we included nine copies of each protein, allowing up to 27 tethers to cooperate in determining the localization of SVs. According to the model and its imposed assumptions, the formation of additional tethers promotes shorter distances between SV and PM (Fig. 2). A similar outcome has been derived from a coarse-grained model where additional SNARE complexes caused shorter SV–PM distances via steric-electrostatic interactions between the SNAREs and the SV and PM (Mostafavi et al., 2017). Correspondingly, in several preparations imaged with EM, SVs located closer to the PM typically were tethered to the PM by multiple filaments (Fernández-Busnadiego et al., 2013; Papantoniou et al., 2023; Szule et al., 2012).

The assumption of up to nine SV–PM tethers per protein was based on experiments using single-molecule imaging of Munc13-1 (Sakamoto et al., 2018) and, for simplicity, we assumed the same number of SNARE and Syt tethers. However, in reality, 70 synaptobrevins and 15 synaptotagmins are estimated to be present per SV (Takamori et al., 2006; Wilhelm et al., 2014). It is uncertain whether all of these can form protein tethers because the availability of t-SNAREs and PI(4,5)P$_2$ clusters on the PM might also be limited. Nonetheless, to simulate a biologically more realistic scenario, we adapted our model to include up to 35 SNARE tethers and seven Syt tethers, considering only half of the proteins on the SV face the PM (Fig. 9). This version of the model revealed smaller median SV–PM distances and a larger proportion of SVs within 5 nm of the PM, consistent with the notion that the SNAREs are responsible to stabilize PM-proximal SVs, and with this being more efficient when their number increases from 9 to 35 (Fig. 9*A–E*). We also saw a tendency to increase the latency of SVs to move within 5 nm of the PM, consistent with the ability to associate fewer Syt tethers which speed up this process (Fig. 9*F* and *G*).

In another version of the model, we also adjusted the association and disassembly rate constants of the SNARE and Syt tethers to achieve the same average number of associated SV tethers at a steady-state as our simplified model with a maximum of nine tethers each. This resulted in very similar SV dynamics and steady-state distributions as in our simplified model where the maximal number of tethers was nine for all three proteins (Fig. 9*H–M*).

This indicates that the maximal number of tethers that can form and their association rate cannot be determined independently in our model. However, it also means that the conclusions obtained from our simplified model can also be extended to biologically more plausible situations where different tethering proteins are available in different numbers. Biologically, it is also highly probable that the maximal rates of SNARE complex assembly, and membrane binding of Munc13 and Syt differ (Hui et al., 2005; Li et al., 2016; Shin et al., 2010; van den Bogaart et al., 2012). Moreover, these rates are affected by the intracellular environment [e.g. DAG and PI(4,5)P$_2$ levels] (Shin et al., 2010; van den Bogaart et al., 2012), which might vary between synapses and depending on the past activity of a synaptic terminal. Variations in tether abundance and/or formation kinetics might explain the functional heterogeneity observed in various synapses (Maus et al., 2020).

Our model simulations suggest that, on average, one to three copies of each protein actively tether the SV to the PM (Fig. 4). Similarly, in other modelling approaches, we recently estimated a relatively low stoichiometry for Munc13 (two) and Syt (three) in stabilizing SV priming and in executing SV fusion (Jusyte et al., 2023; Kobbersmed et al., 2022). Similarly, estimates for the number of SNARE complexes formed for action-potential induced exocytosis ranged between two to six (Arancillo et al., 2013; Mohrmann et al., 2010; Sinha et al., 2011) and cryo-EM imaging reported six protein complexes of docked vesicles (Radhakrishnan et al., 2021). The nature of the cooperation in our model was indirect (tethers formed and acted independently) and contrasts with studies suggesting the formation of large protein oligomers, or rings, including a fixed number of proteins of the same type (e.g. only Syt or Munc13) (Grushin et al., 2022; Radhakrishnan et al., 2021; Rothman et al., 2017; Zhu et al., 2022). This would indirectly imply that, if the number of protein copies is reduced to a level below the stoichiometry of the ring, vesicles would be unable to dock correctly.

## Long protein tethers increase the speed of SV docking

Our model shows that longer protein tethers enhance the speed of SNARE complex formation and promote SVs to move within 5 nm from the PM. This can be interpreted as faster rates of SV docking, when considering SVs within a distance of 5 nm from the PM as *docked*. The rates of docking and undocking determine the speed of recovery of the readily releasable pool after an AP, and thereby play a role during short-term plasticity. Interestingly, the dynamic, Ca$^{2+}$-dependent regulation of docking and/or undocking is assumed to play a prominent role in determining the short-term plasticity characteristics of

a synapse (Chang et al., 2018; Kobbersmed et al., 2020; Kusick et al., 2022; Malagon et al., 2020; Miki et al., 2016; Pan & Zucker, 2009; Pulido & Marty, 2018; Silva et al., 2021; Weingarten et al., 2022). Repetitive stimulation elevates the intracellular $Ca^{2+}$ levels which might recruit additional protein tethers, such as Syt3 and Syt7 (Jackman et al., 2016; Weingarten et al., 2022; Wu et al., 2024). Moreover, the elevated $Ca^{2+}$-levels are assumed to induce changes in the configuration of Munc13 and Syt1/2 tethers (Camacho et al., 2021; Lin et al., 2014; van den Bogaart et al., 2011; Xu et al., 2017). Based on our model results, we expect that the increased number of tethers recruited during repetitive stimulation, along with greater variability in their lengths, will enhance the rate of docking further. This might provide a molecular explanation for the $Ca^{2+}$-dependent docking rates suggested in multiple studies (Kobbersmed et al., 2020; Kusick et al., 2020, 2022; Lin et al., 2022; Malagon et al., 2020; Miki et al., 2018; Pan & Zucker, 2009; Pulido & Marty, 2018; Silva et al., 2021; Weingarten et al., 2022). In the future, the model could be adapted to investigate this hypothesis.

### Limitations and model assumptions

The goal of this model was to extract some general principles of SV docking by using a simplified approach. Because of the assumptions used to construct the model, there are some important limitations to consider:

(1) The model only includes the SNARE complex, Munc13 and Syt. We excluded other proteins that could affect the localization and trafficking of SVs, as has been shown previously for RIM (Fernández-Busnadiego et al., 2013) and Bassoon (Hallermann et al., 2010). The lack of these components could explain the smaller accumulation distance of SVs in our Munc13-, SNARE-condition compared to Cryo-EM results obtained with Munc13 DKOs ($\sim$10 *vs.* $\sim$16 nm) (Papantoniou et al., 2023). We decided not to include these longer protein tethers in our model because there is less structural information available for these proteins.

(2) To calculate the distance at which protein tethers form and where formed tethers keep the SV, we assumed tether formation to occur on the entire lower hemisphere of the SV. This agrees with EM data showing that most proteins tethering the PM to SV are connected seemingly randomly on the SV hemisphere facing the PM (Harlow et al., 2013; Szule et al., 2012). With this assumption, we ignored several aspects constraining the localization of tethers, such as the preferred orientation of domains inserted into the PM or SV membranes and AZ scaffolding proteins. We also did not simulate the spatial relationship between several tethers. The exact location of the

protein tethers does impact the localization of SVs with respect to the PM a lot, as demonstrated in a previous modelling study (Jung & Doniach, 2017). In the future, there is a need to explore further how the distance between SV and PM is affected when including more spatial constraints of individual protein tethers and interactions between tethers.

(3) Our model does not include information on the localization of other synaptic elements or neighbouring SVs that may constrain trafficking. Yet it is well established that tethers not only connect between the SV and the PM, but also in between SVs (Gustafsson et al., 2002; Szule et al., 2012). This will clearly affect SV mobility, but, because our model only includes membrane tethering and the movement of individual SVs, these interactions were not considered.

(4) In our model, we implemented the tethers as independent functional units. Although this facilitates the simulations and indeed accounts for many aspects of SV trafficking to the PM, this simplification ignores some important biological aspects. For example, our model can only approximate Munc13 DKO phenotypes if the loss of Munc13 also impairs tethering via the SNAREs. Although this is consistent with the requirement of Munc13 for SNARE complex formation (Ma et al., 2011; Magdziarek et al., 2020; Wang et al., 2019), it also shows a clear limitation of our model because this biologically relevant interdependence is ignored in the independent implementation of the SNARE and Munc13 tethers. A similar situation holds for the Syt tethers because Syt interacts with the SNAREs and these interactions influence vesicle docking (Mohrmann et al., 2013; Zhou et al., 2015), but we also did not take this interdependence into account.

(5) Because we currently lack experimental information on the assembly and disassembly rates, the model can only be used to simulate steady-state distributions or obtain information on the sequence of events, rather than the exact timing of events. Future experiments using single-molecule approaches may help estimate those rates which would allow us to implement a time-dependent version of our model. This would immediately allow us to link the SV dynamics predicted here to short-term plasticity phenomena which probably rely on fast docking state transitions (Neher, 2024).

Despite these simplifications, the model provides a useful framework to study the effects of tethers on SV trafficking and docking. It also uncovers that long protein tethers enhance SV trafficking to the PM. In our model, the formation of tethers can progress even if some proteins are missing. This is of value in the interpretation of

experimental data obtained using, for example, EM after the (genetic) removal of certain proteins, and allows for the quick investigation of new hypotheses. With this, future extensions and explorations of the model can provide a more holistic understanding of how proteins cooperate during SV docking.

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

## Additional information

### Data availability statement

All figure source data and software codes used in the study are available upon request from Manon Berns (manon.berns@sund.ku.dk) or Alexander Walter (awalter@sund.ku.dk).

### Competing interests

The authors declare that they have no competing interests.

### Author contributions

Experiments were performed in the laboratory of Alexander M. Walter, Molecular and Theoretical Neuroscience, Department of Neuroscience, Faculty of Health and Medical Sciences, University of Copenhagen, and in the research group of Stefanie Winkelmann, Computational Systems Biology, Department of Modelling and Simulation of Complex Processes, Zuse Institute Berlin. M.M.M.B., A.M.W. and S.W. were responsible for the conception or design of the work. M.M.M.B., A.M.W., S.W. and M.Y. were responsible for acquisition, analysis or interpretation of data. M.M.M.B., A.M.W., M.Y. and S.W. were responsible for drafting or revising the work. All authors approved the final version of the manuscript submitted for publication, and agree to be accountable for all aspects of the work in ensuring that questions related to the accuracy or integrity of any part of the work are appropriately investigated and resolved. All persons designated as authors qualify for authorship, and all those who qualify for authorship are listed.

### Funding

This work was funded by a Novo Nordisk Foundation Young Investigator Award to Alexander M. Walter (NNF19OC0056047). We further acknowledge the support provided by Deutsche Forschungsgemeinschaft (DFG) through grant CRC 1114 (Project No. 235221301) and through Germany's Excellence Strategy MATH+: Berlin Mathematics Research Centre (EXC 2046/1, Project No. 390685689).

### Acknowledgements

We thank Cordelia Imig, Anna Schrøder Lassen and Jakob Balslev Sørensen for helpful discussions and feedback.

### Keywords

mathematical modelling, protein tethers, synaptic vesicle, vesicle docking

## Supporting information

Additional supporting information can be found online in the Supporting Information section at the end of the HTML view of the article. Supporting information files available:

**Peer Review History**

