## [Peer Review History · The Journal of Physiology]

Independently engaging protein tethers of different length enhance synaptic vesicle trafficking to the plasma membrane

Alexander Matthias Walter, Manon M.M. Berns, Stefanie Winkelmann, and Mirza Yildiz

DOI: 10.1113/JP286651

Corresponding author(s): Alexander Walter (awalter@sund.ku.dk)

Review Timeline:

Submission Date:	03-Apr-2024
Editorial Decision:	07-May-2024
Revision Received:	14-Nov-2024
Accepted:	18-Dec-2024

Senior Editor: Katalin Toth

Reviewing Editor: Samuel Young

Transaction Report:

Dear Dr Walter,

Re: JP-RP-2024-286651 "Sequential assembly of longer to shorter protein tethers enhances the speed of synaptic vesicle trafficking toward the plasma membrane" by Alexander Matthias Walter and Manon M.M. Berns

Thank you for submitting your manuscript to The Journal of Physiology. It has been assessed by a Reviewing Editor and by 2 expert referees and we are pleased to tell you that it is potentially acceptable for publication following satisfactory major revision.

REVISION CHECKLIST:

We look forward to receiving your revised submission.

Yours sincerely,

Katalin Toth
Senior Editor
The Journal of Physiology

REQUIRED ITEMS FOR REVISED SUBMISSION

- Author photo and profile. First or joint first authors are asked to provide a short biography (no more than 100 words for one author or 150 words in total for joint first authors) and a portrait photograph. These should be uploaded and clearly labelled together in a Word document with the revised version of the manuscript. See Information for Authors for further details.

- Please upload separate high-quality figure files via the submission form.

- Please ensure that the Article File you upload is a Word file.

- Your paper contains Supporting Information of a type that we no longer publish, including supplementary tables and figures. Any information essential to an understanding of the paper must be included as part of the main manuscript and figures. The only Supporting Information that we publish are video and audio, 3D structures, program codes and large data files. Your revised paper will be returned to you if it does not adhere to our Supporting Information Guidelines.

- Please include an Abstract Figure file, as well as the Figure Legend text within the main article file. The Abstract Figure is a piece of artwork designed to give readers an immediate understanding of the research and should summarise the main conclusions. If possible, the image should be easily 'readable' from left to right or top to bottom. It should show the physiological relevance of the manuscript so readers can assess the importance and content of its findings. Abstract Figures should not merely recapitulate other figures in the manuscript. Please try to keep the diagram as simple as possible and without superfluous information that may distract from the main conclusion(s). Abstract Figures must be provided by authors no later than the revised manuscript stage and should be uploaded as a separate file during online submission labelled as File Type 'Abstract Figure'. Please also ensure that you include the figure legend in the main article file. All Abstract Figures should be created using BioRender. Authors should use The Journal's premium BioRender account to export high-resolution images. Details on how to use and access the premium account are included as part of this email.

REVIEWING EDITOR COMMENTS

This manuscript developed a mathematical model to evaluate how tethers impact synaptic vesicle fusion. Both reviewers found the topic interesting and important. However, both reviews felt that model was speculative and simplistic. In addition, reviewer#1 pointed out that the authors do not consider previously published experimental data evaluated how increasing tethering lengths in Syb2 impact SV fusion. Therefore, given these concerns the authors should perform an experiment that manipulates tethering lengths to test their model or an alternative would be to use the previously published Syb2 data to test their model. In addition, the authors should work to improve their model to be more representative of the biological landscape in the presynaptic terminal, i.e. different relative copy numbers of proteins.

REFEREE COMMENTS

Referee #1:

In this study, the authors have developed a mathematical model that explains synaptic vesicle trafficking toward the plasma membrane for the vesicle fusion. The authors employed three proteins (synaptotagmin, SNARE proteins, and Munc13) that are involved in tethering of synaptic vesicles to the active zone in the plasma membrane. According to the model, sequential assembly of longer to shorter protein tethers (from synaptotagmin, Munc13 to the SNARE assembly) enhanced the speed of synaptic vesicle trafficking toward the plasma membrane. The authors posit that the predictions of the model are consistent with the synaptic vesicle-plasma membrane distribution results obtained by employing electron microscopy (EM).

Major concern:

1. The authors proposed that sequential assembly of longer to shorter protein tethers enhances the speed of synaptic vesicle trafficking toward the plasma membrane based on their model predictions. The authors mentioned that their model prediction aligned with SV-PM distribution results that are obtained by employing EM. However, the snapshots of EM do not usually suggest any changes of the speed of synaptic vesicle trafficking. This shortcoming diminishes the impact of this paper.

2. Earlier studies (PMID: 16793874, PMID: 17956735) have experimental testing the impact of increasing tethering lengths of synaptobrevin2 on release. This work largely proposed changes in calcium dependence of release as well as differences in tethering length dependence of spontaneous versus evoked release. This earlier is not discussed nor taken into account in the current analysis.

Minor concerns:

The authors may want to improve their model further, but at this point, the model is developed based on too simple assumptions:

1. Unlike the authors' assumption, synaptotagmin and Munc13 are interacting with the SNARE proteins.
2. The copy number of Syb2 is ~5 times more than that of synaptotagmin1 in synaptic vesicles - the difference is too large to ignore in the model.
3. The affinities of the tethering of synaptotagmin, Munc13, and the SNARE proteins are probably quite different.

Referee #2:

In this valuable theoretical study Berns and Walter report on the consequences of long and short tethers on synaptic vesicle trafficking to the plasma membrane and loose and tight docking gives some useful new insights on the putative role of tethers of different lengths as observed in cryo-EM pictures and tomograms of synapses.

They investigated the importance of tethers with different lengths during SV docking by constructing a mathematical stochastic model. Using such Monte Carlo simulations, they find that the different lengths and numbers of putative tether proteins at the plasma membrane like the SNARE proteins, Munc13 or synaptotagmin determine both the rate and tightness of synaptic vesicle tethering to the plasma membrane. They also used this model to study qualitatively the expected effects when a certain class of tether proteins has been removed genetically. They can indeed show that such simulations qualitatively align reasonably well with published ultrastructural data, in which either protein type has been genetically ablated.

The study, however, is highly hypothetical and speculative in nature. Yet, this paper would be of clear interest to cell physiologists and biophysicists in the field.

I recommend minor revision of the manuscript before publication in the Journal of Physiology:

- In equation 11 on the right-hand side the index of b should be j prime instead of j.

- The authors state that they use uniformly distributed random numbers. They should state, which algorithm they used (MatLab built-in uniform random number generator?)

- When discussing the limitations of the model in the Discussion section, the authors mention, that a possible impact of neighboring vesicles has been dismissed in their model. In cryo-EM pictures and tomograms, however, vesicles clearly are interconnected by numerous tethers of different length, which indeed should highly constrain the movement of vesicle towards the plasma membrane. They authors should discuss this point.

END OF COMMENTS

Response to Referees

We thank the reviewers and reviewing editors for evaluating our work and for their constructive feedback for further improving our manuscript. Based on this input, we revised our manuscript by major changes to text and figure items and by including additional simulations. We hope that with these improvements our resubmitted manuscript meets all criteria for publication at the Journal of Physiology. Below please find our point-by-point response.

REVIEWING EDITOR COMMENTS

This manuscript developed a mathematical model to evaluate how tethers impact synaptic vesicle fusion. Both reviewers found the topic interesting and important. However, both reviews felt that model was speculative and simplistic. In addition, reviewer#1 pointed out that the authors do not consider previously published experimental data evaluated how increasing tethering lengths in Syb2 impact SV fusion. Therefore, given these concerns the authors should perform an experiment that manipulates tethering lengths to test their model or an alternative would be to use the previously published Syb2 data to test their model. In addition, the authors should work to improve their model to be more representative of the biological landscape in the presynaptic terminal, i.e. different relative copy numbers of proteins.

Response: We thank the reviewing editor for this assessment and for clearly pointing out the weaknesses and necessary improvements to our paper. We have addressed the two points that were raised: analysis of Syb2 elongation and implementation of different relative copy numbers by new simulations and additional manuscript figures (new Figure 3 and Figure 9).

We find that extending the tethering length for Syb2 leads to a shift to larger distances between the SV and the plasma membrane (new Figure 3C,D). Most notably, the population of vesicles at a distance below 5nm – often attributed to the primed SVs of the readily releasable pool (RRP) – was markedly reduced (Figure 3E). While this is in qualitative agreement with the experiments by the Bruns and Sudhof lab that found a decreased RRP size (Deák *et al.*, 2006; Kesavan *et al.*, 2007), we also note that the physiological consequences of tether elongation may be larger due to effects on the SV fusion reaction which our model cannot predict. We have discussed this limitation in the text.

We adapted the model to include more realistic protein stoichiometries for the tethering proteins. We included new simulations with adjusted stoichiometries based on published experiments: In addition to the super resolution imaging study by Sakamoto and colleagues based on which we assumed 9 Munc13 proteins to be present (Sakamoto *et al.*, 2018), we considered the biochemical data by the Jahn and Rizzoli labs that had found 70

copies of Syb2 and 15 copies of Syt1 (!!! INVALID CITATION !!! (Takamori et al., 2006; Wilhelm et al., 2014)). We adjusted the maximal number of SNARE tethers and Syt tethers in this version of the model to each 50% of these values, 35 Syb2 and 7 Syt, assuming that only proteins at the lower hemisphere of the SV can interact with the PM. These results are shown in the new Figure 9 where they are compared to our simplified model with each 9 tethers. With these biologically more realistic stoichiometries we see that SVs occupy shorter distances and the number of SVs within 5nm of the plasma membrane increases provided we keep the association/dissociation rate constants the same as in our simplified model with max 9 SNARE/Syt each (Figure 9A-G).

We also derived an expression to adjust the association and dissociation rate constants in dependence on the protein copy number that would sustain the same steady state number of associated tethers as in our simplified model with only up to 9 tethers each. Under these conditions we see overall very similar steady state behavior for both versions of our model, indicating that the model per se can be used to simulate biologically more plausible situations (Figure 9H-M). We note that to achieve this agreement with more SNARE proteins, the maximal rate constant of tether assembly was proportionally reduced and the rate of tether disassembly slightly increased. In contrast, we implemented fewer Syt tethers by proportionally increasing the maximal tether assembly rate constant and by slightly reducing the dissociation rate constant.

Responses to Referee #1:

In this study, the authors have developed a mathematical model that explains synaptic vesicle trafficking toward the plasma membrane for the vesicle fusion. The authors employed three proteins (synaptotagmin, SNARE proteins, and Munc13) that are involved in tethering of synaptic vesicles to the active zone in the plasma membrane. According to the model, sequential assembly of longer to shorter protein tethers (from synaptotagmin, Munc13 to the SNARE assembly) enhanced the speed of synaptic vesicle trafficking toward the plasma membrane. The authors posit that the predictions of the model are consistent with the synaptic vesicle-plasma membrane distribution results obtained by employing electron microscopy (EM).

Major concern:

1. The authors proposed that sequential assembly of longer to shorter protein tethers enhances the speed of synaptic vesicle trafficking toward the plasma membrane based on their model predictions. The authors mentioned that their model prediction aligned with SV-PM distribution results that are obtained by employing EM. However, the snapshots of EM do not usually suggest any changes of the speed of synaptic vesicle trafficking. This shortcoming diminishes the impact of this paper.

Response: This reviewer is raising the highly relevant point in the sense that kinetic data cannot be obtained from the EM studies capturing resting SV distributions that we have used to estimate some of the parameters of our model from. We have re-worked the text of our manuscript to clearly state this challenge. We point out clearly that since we are comparing steady state (resting) SV distributions, we can only estimate the ratio between tether association and dissociation rates. This also becomes clear from the parameter sensitivity analysis now shown in the main Figure 6. This means that we cannot use our model to predict the absolute timing of events, but we can infer the sequence of reactions, which we now point out more clearly. e.g. by this added sentence here:

“While this demonstrates that this model can be useful for extracting steady state features of SV docking, we would once again like to point out that the dynamic aspects suffer from the lack of knowledge of these rate constants. Therefore, the information we provide regarding timing needs to be taken with caution and rather interpreted regarding the sequence of reactions than their absolute timing.”

Because of this limitation of the model, we have also adjusted the title of our manuscript and removed the term “speed” which the reviewer rightfully states we cannot determine in absolute terms. We still think even without absolute information on time, assessing the temporal relation of reactions is useful, and our model provides access to this.

2. Earlier studies (PMID: 16793874, PMID: 17956735) have experimental testing the impact

of increasing tethering lengths of synaptobrevin2 on release. This work largely proposed changes in calcium dependence of release as well as differences in tethering length dependence of spontaneous versus evoked release. This earlier is not discussed nor taken into account in the current analysis.

Response: We have now cited and discussed these papers, and we have included additional simulations to test the effects induced by such modification in our model. We thank the reviewer for this suggestion.

We had not considered these studies before, due to the limitation that our model can only be used to simulate SV docking at resting synapses, but not SV fusion. But indeed, both studies mentioned by the reviewer reported on a decreased size of the readily releasable pool (RRP) upon extension of the linker between the Synaptobrevin2 transmembrane domain and its SNARE motif. And since the RRP is dependent on SV docking, evaluating these modifications in our model is an excellent idea. We therefore included additional simulations now included in a new Figure 3 in our revised manuscript where the 'effective height' of the SNARE tether was increased by 5 nm and 10 nm (Figure 3B). We found that this resulted in a progressive increase in the median SV-PM distance (Figure 3C). Interestingly, the proportion of SVs found within 5nm of the PM was markedly reduced (Figure 3E), and these SVs are often considered to constitute the RRP. This finding is therefore in qualitative agreement with the decrease in RRP size reported in the mentioned studies.

We also discuss that the changes seen on the SV-PM distance distribution in our simulations upon SNARE tether extension are less pronounced than the physiological effects seen experimentally. We interpret this as additional effects on the SV fusion reaction which we cannot describe with our model. We therefore agree with the reviewer that this points towards relevant effects of linker extension on the Ca²⁺-induced SV fusion reaction and describe this in our text.

Minor concerns:

The authors may want to improve their model further, but at this point, the model is developed based on too simple assumptions:

1. Unlike the authors' assumption, synaptotagmin and Munc13 are interacting with the SNARE proteins.

Response: This is true. And we have made sure that this simplification is clearly stated in our manuscript text. Indeed, our model further supports the relevance of functional interaction between Munc13 and the SNARE proteins because our analysis of Munc13KO points to the fact that removal of the Munc13-tethering function alone is not sufficient to explain the dramatic SV docking phenotype seen experimentally. Our analysis suggests that this phenotype is best explained by an additional disruption of the SNARE tethers (in what we call the Munc13 -, SNARE - condition). This clearly shows that functional

interdependence between these tethering proteins is biologically relevant. We have made sure to state this clearly in our text:

“We therefore conclude that other than assumed in our simplified model of independent tether action, tethers functionally interact, and this interaction is biologically relevant.”

We have also included the following text on Munc13 and Syt in the section describing the limitations of our study:

“In our model we implemented the tethers as independent functional units. While this facilitates the simulations and indeed accounts for many aspects of SV trafficking to the PM, this simplification ignores some important biological aspects. For instance, our model can only approximate Munc13 DKO phenotypes if loss of Munc13 also impairs tethering via the SNAREs. While this is consistent with the requirement of Munc13 for SNARE complex formation (Ma et al., 2011; Wang et al., 2019; Magdziarek et al., 2020), it also shows a clear limitation of our model because this biologically relevant inter-dependence is ignored in the independent implementation of the SNARE and Munc13 tethers. A similar situation holds for the Syt tethers, because Syt interacts with the SNAREs and these interactions influence vesicle docking (Mohrmann et al., 2013; Zhou et al., 2015) but we also did not take this inter-dependence into account.”

2. The copy number of Syb2 is ~5 times more than that of synaptotagmin1 in synaptic vesicles - the difference is too large to ignore in the model.

Response: That is correct. We have now included new simulations shown in Figure 9 of the revised manuscript to take this into account. In this version of the model, we considered 70 Syb2 proteins and 15 Syt proteins, based on biochemical estimates from the Jahn and Rizzoli labs (Takamori et al., 2006; Wilhelm et al., 2014). We have assumed that each 50% of these protein copies would be on the lower hemisphere of the SV and therefore accessible for tether formation. Therefore, this version of the model features 35 SNARE tethers, 9 Munc13 tethers and 7 Syt tethers. We show that this biologically more plausible model predicts a higher population of SVs close to the PM, consistent with more efficient SNARE-dependent docking (Figure 9A-G).

We also added simulations in which we changed the maximal number of tethers but simultaneously adjusted the tether association and dissociation kinetics to achieve a similar steady state binding as in our model with each 9 tethers (Figure 9H-M). More specifically, the increased number of SNAREs was compensated in our model by proportionally decreasing the maximal tether association rate constant and slightly increasing the dissociation rate constant. The opposite changes were applied to the Syt to account for the slightly lower copy number compared to our control model (7 instead of 9). Simulations of this model produced very similar SV dynamics and steady state distributions as our simplified model with only 9 tethers each. While this demonstrates that

we cannot distinguish the number of tethers independently from their binding kinetics, it also indicates that the conclusions drawn from our simplified 9-tether model can be extended to biologically more plausible tether numbers.

3. The affinities of the tethering of synaptotagmin, Munc13, and the SNARE proteins are probably quite different.

Response: Yes, that is true given the different interaction domains and binding partners. This constitutes a strong simplification in our model. We made sure this is clearly pointed out by adding the following text:

“The maximum rate of tether assembly and the rate of tether disassembly are unknown. We started out by using the same assembly rates for all tethers. This simplification ignores that the interaction domains of the three tethering proteins are different which is expected to result in markedly different rates.”

The rates also depend on the protein stoichiometry and the newly added simulations including more realistic protein copy numbers (Figure 9H-M) demonstrate this where we predict different on- and off rates for all three protein types (see Table 1).

Referee #2:

In this valuable theoretical study Berns and Walter report on the consequences of long and short tethers on synaptic vesicle trafficking to the plasma membrane and loose and tight docking gives some useful new insights on the putative role of tethers of different lengths as observed in cryo-EM pictures and tomograms of synapses.

They investigated the importance of tethers with different lengths during SV docking by constructing a mathematical stochastic model. Using such Monte Carlo simulations, they find that the different lengths and numbers of putative tether proteins at the plasma membrane like the SNARE proteins, Munc13 or synaptotagmin determine both the rate and tightness of synaptic vesicle tethering to the plasma membrane. They also used this model to study qualitatively the expected effects when a certain class of tether proteins has been removed genetically. They can indeed show that such simulations qualitatively align reasonably well with published ultrastructural data, in which either protein type has been genetically ablated.

The study, however, is highly hypothetical and speculative in nature. Yet, this paper would be of clear interest to cell physiologists and biophysicists in the field.

I recommend minor revision of the manuscript before publication in the Journal of Physiology:

Response: We thank the reviewer for this assessment

- *In equation 11 on the right-hand side the index of b should be j prime instead of j.*

Response: Thank you for pointing this out. We corrected this mistake.

- *The authors state that they use uniformly distributed random numbers. They should state, which algorithm they used (MatLab built-in uniform random number generator?)*

Response: Indeed, it was the built in “rand” function. We added this information to the text. Thank you for pointing this out.

- *When discussing the limitations of the model in the Discussion section, the authors mention, that a possible impact of neighboring vesicles has been dismissed in their model. In cryo-EM pictures and tomograms, however, vesicles clearly are interconnected by numerous tethers of different length, which indeed should highly constrain the movement*

of vesicle towards the plasma membrane. They authors should discuss this point.

Response: Thank you for this suggestion, we added this point to the discussion.

References

Deák F, Shin OH, Kavalali ET & Südhof TC. (2006). Structural determinants of synaptobrevin 2 function in synaptic vesicle fusion. *J Neurosci* **26**, 6668-6676.

Kesavan J, Borisovska M & Bruns D. (2007). v-SNARE actions during Ca(2+)-triggered exocytosis. *Cell* **131**, 351-363.

Ma C, Li W, Xu Y & Rizo J. (2011). Munc13 mediates the transition from the closed syntaxin-Munc18 complex to the SNARE complex. *Nat Struct Mol Biol* **18**, 542-549.

Magdziarek M, Bolembach AA, Stepien KP, Quade B, Liu X & Rizo J. (2020). Re-examining how Munc13-1 facilitates opening of syntaxin-1. *Protein Sci* **29**, 1440-1458.

Mohrmann R, de Wit H, Connell E, Pinheiro PS, Leese C, Bruns D, Davletov B, Verhage M & Sorensen JB. (2013). Synaptotagmin interaction with SNAP-25 governs vesicle docking, priming, and fusion triggering. *J Neurosci* **33**, 14417-14430.

Sakamoto H, Ariyoshi T, Kimpara N, Sugao K, Taiko I, Takikawa K, Asanuma D, Namiki S & Hirose K. (2018). Synaptic weight set by Munc13-1 supramolecular assemblies. *Nat Neurosci* **21**, 41-49.

Takamori S, Holt M, Stenius K, Lemke EA, Grønborg M, Riedel D, Urlaub H, Schenck S, Brügger B, Ringler P, Müller SA, Rammner B, Gräter F, Hub JS, De Groot BL, Mieskes G, Moriyama Y, Klingauf J, Grubmüller H, Heuser J, Wieland F & Jahn R. (2006). Molecular anatomy of a trafficking organelle. *Cell* **127**, 831-846.

Wang S, Li Y, Gong J, Ye S, Yang X, Zhang R & Ma C. (2019). Munc18 and Munc13 serve as a functional template to orchestrate neuronal SNARE complex assembly. *Nat Commun* **10**, 69.

Wilhelm BG, Mandad S, Truckenbrodt S, Kröhnert K, Schäfer C, Rammner B, Koo SJ, Claßen GA, Krauss M, Haucke V, Urlaub H & Rizzoli SO. (2014). Composition of isolated synaptic boutons reveals the amounts of vesicle trafficking proteins. *Science* **344**, 1023-1028.

Zhou Q, Lai Y, Bacaj T, Zhao M, Lyubimov AY, Uervirojnangkoorn M, Zeldin OB, Brewster AS, Sauter NK, Cohen AE, Soltis SM, Alonso-Mori R, Chollet M, Lemke HT, Pfuetzner RA,

Choi UB, Weis WI, Diao J, Sudhof TC & Brunger AT. (2015). Architecture of the synaptotagmin-SNARE machinery for neuronal exocytosis. *Nature* **525**, 62-67.

Dear Dr Walter,

Re: JP-RP-2024-286651R1 "Independently engaging protein tethers of different length enhance synaptic vesicle trafficking to the plasma membrane" by Alexander Matthias Walter, Manon M.M. Berns, Stefanie Winkelmann, and Mirza Yildiz

We are pleased to tell you that your paper has been accepted for publication in The Journal of Physiology.

Yours sincerely,

Katalin Toth
Senior Editor
The Journal of Physiology

If you would like to receive our 'Research Roundup', a monthly newsletter highlighting the cutting-edge research published in The Physiological Society's family of journals (The Journal of Physiology, Experimental Physiology, Physiological Reports, The Journal of Nutritional Physiology and The Journal of Precision Medicine: Health and Disease), please click this link, fill in your name and email address and select 'Research Roundup':
<https://www.physoc.org/journals-and-media/membernews>

- You can help your research get the attention it deserves! Check out Wiley's free Promotion Guide for best-practice recommendations for promoting your work at: www.wileyauthors.com/eeo/guide. You can learn more about Wiley Editing Services which offers professional video, design, and writing services to create shareable video abstracts, infographics, conference posters, lay summaries, and research news stories for your research at: www.wileyauthors.com/eeo/promotion.

EDITOR COMMENTS

Reviewing Editor:

The authors have done an excellent job responding to the reviewers comments. There are no further concerns.

REFeree COMMENTS

Referee #1:

The authors have addressed my earlier comments and improved the manuscript.

Referee #2:

I am satisfied with the authors' response to my comments. I have no further concerns or comments.